# Stochastic Zeroth-Order Optimization under Strongly Convexity and Lipschitz Hessian: Minimax Sample Complexity

**Qian Yu**
University of California, Santa Barbara
qianyu02@ucsb.edu

**Yining Wang**
University of Texas at Dallas
yining.wang@utdallas.edu

**Baihe Huang**
University of California, Berkeley
baihe_huang@berkeley.edu

**Qi Lei**
New York University
ql518@nyu.edu

**Jason D. Lee**
Princeton University
Jasondl@princeton.edu

## Abstract

Optimization of convex functions under stochastic zeroth-order feedback has been a major and challenging question in online learning. In this work, we consider the problem of optimizing second-order smooth and strongly convex functions where the algorithm is only accessible to noisy evaluations of the objective function it queries. We provide the first tight characterization for the rate of the minimax simple regret by developing matching upper and lower bounds. We propose an algorithm that features a combination of a bootstrapping stage and a mirror-descent stage. Our main technical innovation consists of a sharp characterization for the spherical-sampling gradient estimator under higher-order smoothness conditions, which allows the algorithm to optimally balance the bias-variance tradeoff, and a new iterative method for the bootstrapping stage, which maintains the performance for unbounded Hessian.

## 1  Introduction

Stochastic optimization of an unknown function with access to only noisy function evaluations is a fundamental problem in operations research, optimization, simulation and bandit optimization research, commonly known as *zeroth-order optimization* (Chen et al., 2017), *derivative-free optimization* (Conn et al., 2009; Rios & Sahinidis, 2013) or *bandit optimization* (Bubeck et al., 2021). In this problem, an optimization algorithm interacts sequentially with an oracle and obtains noisy function evaluations at queried points every time. The algorithm produces an approximately optimal solution after $T$ such evaluations, with its performance evaluated by the expected difference between the function values at the approximate optimal solution produced and the optimal solution. A more rigorous formulation of the problem is given in Sec. 2 below.

Existing works and results on stochastic zeroth-order optimization could be broadly categorized into two classes:

1. **Convex functions**. In the first thread of research, the unknown objective function to be optimized is assumed to be *concave* (for maximization problems) or *convex* (for minimization problems). For these problems, with minimal smoothness (e.g. objective function being Lipschitz continuous) it is possible to achieve a sample complexity of $\tilde{O}(\varepsilon^{-2})$ for an expected optimization error or $\varepsilon$, which is also a polynomial function of domain dimension

38th Conference on Neural Information Processing Systems (NeurIPS 2024).

| Lower Bound | Upper Bounds | |
|---|---|---|
| | Bach & Perchet (2016) $O(dT^{-\frac{1}{2}}M^{-\frac{1}{2}})$ | Akhavan et al. (2020) $O(d^2T^{-\frac{2}{3}}M^{-1})$ |
| $\mathbf{\Omega(dT^{-\frac{2}{3}}M^{-1})}$ | Novitskii & Gasnikov (2021) $O(d^{\frac{5}{3}}T^{-\frac{2}{3}}M^{-1})$ | **Ours** $\mathbf{O(dT^{-\frac{2}{3}}M^{-1})}$ |

Table 1: The dependence of simple regret on $T$ (number of function evaluations), $d$ (dimension) and $M$ (parameter describing strong convexity). Our results are highlighted in comparison to the prior works.

$d$; see for example the works of Agarwal et al. (2013); Lattimore & Gyorgy (2021); Bubeck et al. (2021);

2. **Smooth functions**. In the second thread of research, the unknown objective function to be optimized is assumed to be highly *smooth*, but not necessary concave/convex. Typical results assume the objective function is Hölder smooth of order $k \geq 1$, meaning that the $(k-1)$-th derivative of the objective function is Lipschitz continuous. Without additional conditions, the optimal sample complexity with such smoothness assumptions is $\tilde{O}(\varepsilon^{-(2+d/k)})$ (Wang et al., 2019), which scales exponentially with the domain dimension $d$.

In this paper, we study the optimal sample complexity of stochastic zeroth-order optimization when the objective function exhibits both (strong) convexity and a high degree of smoothness. As we have remarked in the first bullet point above, with convexity and Hölder smoothness of order $k = 1$ (equivalent to the objective function being Lipschitz continuous), the works of Agarwal et al. (2013); Lattimore & Gyorgy (2021); Bubeck et al. (2021) established an $\tilde{O}(\varepsilon^{-2})$ upper bound. With higher order of Hölder smoothness, i.e., $k = 2$ (equivalent to the gradient of the objective being Lipschitz continuous), it is shown that simpler algorithms exist but the sample complexity remains $\tilde{O}(\varepsilon^{-2})$ (Besbes et al., 2015; Agarwal et al., 2010; Hazan & Levy, 2014), which seemingly suggests the relatively smaller role smoothness plays in the presence of convexity. In this paper we show that with even higher order of Hölder smoothness, i.e., $k = 3$ (specifically, the Hessian of the objective being Lipschitz continuous), the optimal sample complexity is improved to $O(\varepsilon^{-1.5})$, which is significantly smaller than the sample complexity of the convex-without-smoothness setting $\tilde{O}(\varepsilon^{-2})$, or the smooth-without-convexity setting $\tilde{O}(\varepsilon^{-(2+d/3)})$. More importantly, when the Lipschitzness of Hessian is defined in Frobenius norm (see condition A1), we propose an algorithm that also achieves the optimal dimension dependency, which fully characterizes the optimal sample complexity.

**Summary of technical contributions.** We developed several important techniques in this paper to achieve the optimal sample complexity when the objective function is strongly convex and has Lipschitz Hessian. First, we show that when estimating the gradient under a stochastic environment, even with an unbounded action space, it could be beneficial to sample with non-isotropic distributions (as opposed to conventional standard Gaussian, or uniform distributions on hyperspheres). Second, we present a new approach to analyze the bias and variance of the hyperellipsoid-sampling-based gradient estimators, which enables obtaining sharp bounds with tight constants and strengthens the best-known results in the higher-order smoothness case. Third, we present a two-stage bootstrap-type framework for the algorithmic design, which extends the perturbative analysis in the final stage to the full regime. This extension relies on a non-trivial modification of Newton's method, and we proved its robustness under stochastic observation. We complete the characterization of the minimax regret by deriving a lower bound using the KL-divergence-based approach.

**Additional related works on higher-order smoothness.** Recent years have seen increasing attention on exploiting higher order smoothness in bandit optimization. Remarkably, it was shown that when the Hölder smoothness condition holds simultaneously for both $k = 2$ and $k = 3$, the optimal sample complexity can be improved to $O(\varepsilon^{-1.5})$. (Akhavan et al., 2020; Novitskii & Gasnikov, 2021). We list our results together with the most relevant work in Table 1. While this line of work also demonstrates the benefit of higher-order smoothness in improving the sample complexity, their setting is related but slightly different from what we considered in this work. (See reference therein: Bach & Perchet (2016); Akhavan et al. (2020); Novitskii & Gasnikov (2021)). On one hand, the prior work concentrates on projected gradient-descent-like algorithms, which require a Lipschitz gradient (i.e., the $k = 2$ requirement, and we do not). This additional requirement can not be removed by

simply replacing the gradient steps with Newton's methods, which can lead to unbounded expectation in simple regret in the stochastic case.[1] On the other hand, their results are based on the generalized Hölder condition, which is different from our assumption that the Hessian is Lipschitz in Frobenius norm. Therefore we only emphasize the dependence of $d, T$ and $M$ in Table 1 and omit other parameters. We provide a detailed comparison on the implication of these results in Appendix A.

Our results are also related to a special case discussed in (Shamir, 2013), which shows that for *quadratic* functions it is possible to achieve a sample complexity of $\tilde{O}(\varepsilon^{-1})$. As quadratic functions are infinitely differentiable with bounded derivatives on orders, they are Hölder smooth of any arbitrary order $k \to \infty$, which could be regarded as an extreme of the results established in this paper which only require $k = 3$.

**Related works on gradient estimators.** Gradient estimation serves as a key building block for stochastic zeroth-order optimization algorithms. For instance, a classical one-point estimator was proposed as early as in Flaxman et al. (2005); Blair (1985), where the gradient $\nabla f(\boldsymbol{x})$ is estimated based on empirical measures of $f(\boldsymbol{x} + r\boldsymbol{u})$ for some fixed $r$ and i.i.d. uniformly random $\boldsymbol{u}$ on the unit hypersphere. This was later refined to be two-point estimators, and the sampling distribution of $\boldsymbol{u}$ was generalized to isotropic distributions such as standard Gaussian (e.g., see Agarwal et al. (2010); Bach & Perchet (2016); Zhang et al. (2020)). A majority of prior work focused on the analysis for such estimators under the Lipschitz gradient assumption, where the best guaranteed bound for the bias is at the order of $\Theta(r)$, with a polynomial factor dependent on $d$. The line of works by Bach & Perchet (2016); Akhavan et al. (2020); Novitskii & Gasnikov (2021) also adopted isotropic sampling, and it was shown that with higher-order smoothness of $k = 3$, this bound can be improved to $\Theta(r^2)$. The improvement of sample complexity in our work is mainly due to the tight characterization of our gradient estimator, which covers the special case of isotropic sampling and provides a bound of $\frac{r^2 \rho \sqrt{d}}{2(d+2)}$ in the estimation bias. This strengthens or improves the bounds presented in prior works, and a detailed comparison can be found in Appendix A.

On the other hand, non-isotropic sampling was used as early as in Abernethy et al. (2008), then extended in Saha & Tewari (2011); Hazan & Levy (2014). Primarily, they were used to ensure that the sampling points are contained within a bounded action set. (Suggala et al., 2021) showed the necessity of non-isotropic sampling over quadratic loss function in the adversarial setting. In this work, we essentially demonstrated that non-isotropic sampling can be used to refine a preliminary algorithm by adding a mirror-descent-like final stage. More recently, non-isotropic sampling was also adopted in Lattimore & György (2023) to optimize convex and global Lipschitz functions.

**Notations.** We follow the convention of machine learning theory where $\nabla^2 f(\boldsymbol{x})$ denotes the Hessian of $f$ at point $\boldsymbol{x}$, while the trace of Hessian is denoted by $\text{Tr}\left(\nabla^2 f(\boldsymbol{x})\right)$. This should not be confused with the notation in classical field theory, where $\nabla^2 f(\boldsymbol{x})$ instead denotes the trace of the Hessian. We use $\|\cdot\|_2$ to denote vector $\ell_2$ norms, and $\|\cdot\|_F$ to denote matrix Frobenius norms. We use $I_d$ to denote the identity matrix, and $S^{d-1}$ to denote the unit hypersphere centered at the origin, both for the $d$-dimensional Euclidean space $\mathbb{R}^d$. We adopt the conventional notations (i.e., $O$, $\Omega$, $o$, and $\omega$) to describe regret bounds in the asymptotic sense with respect to the total number of samples (denoted by $T$).

## 2 Problem Formulation

We consider the stochastic optimization problem under the class of functions that are strongly convex and have Lipschitz Hessian. The goal in this setting is to design learning algorithms to achieve approximately the global minimum of an unknown objective function $f : \mathbb{R}^d \to \mathbb{R}$.

A learning algorithm $\mathcal{A}$ can interact with the function by adaptively sampling their value for $T$ times, and receive noisy observations. At each time $t \in [T]$, the algorithm selects $\boldsymbol{x}_t \in \mathbb{R}^d$, and receives the

---

[1]We note that even in the classical analysis of Newton's method, which assumes zero-error observations, the additional $k = 2$ smoothness condition was adopted to obtain non-trivial complexity bounds (e.g., see Boyd & Vandenberghe (2004), Section 9.5.3), implying the non-trivialness of removing the $k = 2$ smoothness condition. In this work, we provided an analysis for our proposed bootstrapping algorithm, which ensures the achievability of bounded expected regret even with unbounded hessian.

following observation,

$$y_t = f(\boldsymbol{x}_t) + w_t, \tag{1}$$

where $\{w_t\}_{t=1}^T$ are independent random variables with zero mean and bounded variance. Formally, the algorithm can be described by a list of conditional distributions where each $\boldsymbol{x}_t$ is selected based on all historical data $\{\boldsymbol{x}_\tau, y_\tau\}_{\tau<t}$ and the corresponding distribution. Then for any $t$, we assume that $\mathbb{E}[w_t|\{\boldsymbol{x}_\tau, y_\tau\}_{\tau<t}, \boldsymbol{x}_t] = 0$ and $\mathrm{Var}[w_t|\{\boldsymbol{x}_\tau, y_\tau\}_{\tau<t}, \boldsymbol{x}_t] \leq 1$ for any $t$.[2] For simplicity, we also adopt a common assumption that the additive noises are subgaussian, particularly, $\mathbb{P}[|w_t| > s|\{\boldsymbol{x}_\tau, y_\tau\}_{\tau<t}, \boldsymbol{x}_t] \leq 2e^{-s^2}$ for all $s > 0$ and $t \in [T]$. However, the subgaussian assumption can be removed by adopting more sophisticated mean-estimation methods (e.g., see Nemirovskii & Yuom (1983); Jerrum et al. (1986); Alon et al. (1999); Lee & Valiant (2022); Yu et al. (2023a)).

We assume that the objective function $f$ is second-order differentiable. Furthermore, we impose the following conditions.

(A1) (Lipschitz Hessian). There exist a constant $\rho \in (0, +\infty)$ such that for all $\boldsymbol{x}, \boldsymbol{x}' \in \mathbb{R}^d$, it holds that $\|\nabla^2 f(\boldsymbol{x}) - \nabla^2 f(\boldsymbol{x}')\|_\mathrm{F} \leq \rho \|\boldsymbol{x}' - \boldsymbol{x}\|_2$, where $\|\cdot\|_\mathrm{F}$ denotes the Frobenius norm;

(A2) (Strong Convexity). There exists a constant $M \in (0, +\infty)$ such that for any $\boldsymbol{x} \in \mathbb{R}^d$, the minimum eigenvalue of the Hessian $\nabla^2 f(\boldsymbol{x})$ is greater than $M$.

(A3) (Bounded Distance from Initialization to Optimum Point). There exists a constant $R \in (0, +\infty)$ such that the infimum of $f(\boldsymbol{x})$ within the hyperball $\|\boldsymbol{x}\|_2 \leq R$ is identical to the infimum of $f(\boldsymbol{x})$ over the entire $\mathbb{R}^d$.

In the rest of this paper, we let $\mathcal{F}(\rho, M, R)$ denote the set of all second-order differentiable functions that satisfy the above conditions, with corresponding constants given by $\rho, M$, and $R$. We aim to find algorithms to achieve asymptotically the following minimax simple regret, which measures the expected difference of the objective function on $x_T$ and the optimum.

$$\mathfrak{R}(T; \rho, M, R) := \inf_{\mathcal{A}} \sup_{f \in \mathcal{F}(\rho, M, R)} \mathbb{E}\left[f(\boldsymbol{x}_T) - f(\boldsymbol{x}^*)\right],$$

where $\boldsymbol{x}^*$ denotes the global minimum point of $f$.

## 3 Main Results

**Theorem 3.1.** *For any dimension $d$ and constants $\rho, M, R$, the minimax simple regrets are upper bounded by $\limsup_{T\to\infty} \mathfrak{R}(T; \rho, M, R) \cdot T^{\frac{2}{3}} \leq C \cdot \left(\frac{\rho^{\frac{2}{3}}}{M} d\right)$, where $C$ is a universal constant.*

**Theorem 3.2.** *For any fixed dimension $d$ and constants $\rho, M, R$, the minimax simple regrets are lower bounded by $\liminf_{T\to\infty} \mathfrak{R}(T; \rho, M, R) \cdot T^{\frac{2}{3}} \geq C \cdot \left(\frac{\rho^{\frac{2}{3}}}{M} d\right)$ when the additive noises $w_1, ..., w_T$ are standard Gaussian, where $C$ is a universal constant.*

## 4 Proof Ideas for Theorem 3.1

The proposed algorithm operates in two stages (see Algorithm 4). In the first stage, the algorithm uses a small fraction of samples to obtain a rough estimation of the global minimum point. We ensure that the estimation in the first stage is sufficiently accurate with high probability, so that in the following final stage, the objective function can be approximated by a quadratic function and the resulting approximation error can be bounded using tensor analysis.

### 4.1 Key Techniques and The Final Stage

We first present the key steps of our algorithm, which relies on the subroutines presented in Algorithm 1-3, i.e., GradientEst, BootstrappingEst, and HessianEst. These subroutines estimate the (linearly

---

[2]If the variances of $w_t$'s are bounded by a different constant, all our results can be reproduced by normalizing the values of $f$.

transformed) gradients and Hessian functions of $f$ at any given point by sampling the values of $f$ on hyperellipsoids. The key ingredient of our proof is the sharp characterizations for the biases and variances of the GradientEst estimator, stated in Theorem 4.1.

---

**Algorithm 1** GradientEst

---

**Input**: $\boldsymbol{x}, Z, n$         $\triangleright Z$ is a $d \times d$ matrix, return $\hat{\boldsymbol{g}}$ as an estimator of $Z\nabla f(\boldsymbol{x})$
**for** $k \leftarrow 1$ to $n$ **do**
     Let $\boldsymbol{u}_k$ be a point sampled uniformly randomly from the standard hypersphere $S^{d-1}$
     Let $y_+, y_-$ be samples of $f$ at $\boldsymbol{x} + Z\boldsymbol{u}_k$ and $\boldsymbol{x} - Z\boldsymbol{u}_k$, respectively, let $\boldsymbol{g}_k = \frac{d}{2}(y_+ - y_-)\boldsymbol{u}_k$
**end for**
**Return** $\hat{\boldsymbol{g}} = \frac{1}{n} \sum_{k=1}^{n} \boldsymbol{g}_k$

---

---

**Algorithm 2** BootstrappingEst

---

**Input**: $\boldsymbol{x}, r, n$         $\triangleright$ Goal: estimate $\nabla f(\boldsymbol{x})$ coordinate wise with $O(nd)$ samples
Let $\boldsymbol{e}_1, ..., \boldsymbol{e}_d$ be any orthonormal basis of $\mathbb{R}^d$
**for** $k \leftarrow 1$ to $d$ **do**
     Let $y_{+,k}, y_{-,k}$ each be the average of $n$ samples of $f$ at $\boldsymbol{x} + r\boldsymbol{e}_k$ and $\boldsymbol{x} - r\boldsymbol{e}_k$ respectively
     Let $m_k = (y_+ - y_-)/2r$         $\triangleright$ Estimate the $k$th entry
**end for**
**Return** $\hat{\boldsymbol{m}} = \{m_k\}_{k \in [d]}$

---

---

**Algorithm 3** HessianEst

---

**Input**: $\boldsymbol{x}, r, n$         $\triangleright$ Goal: estimate $\nabla^2 f(\boldsymbol{x})$ coordinate wise with $O(nd^2)$ samples
Let $\boldsymbol{e}_1, ..., \boldsymbol{e}_d$ be any orthonormal basis of $\mathbb{R}^d$
Let $y$ be the average of $n$ samples of $f$ at $\boldsymbol{x}$
**for** $k \leftarrow 1$ to $d$ **do**
     Let $y_{+,k}, y_{-,k}$ each be the average of $n$ samples of $f$ at $\boldsymbol{x} + r\boldsymbol{e}_k$ and $\boldsymbol{x} - r\boldsymbol{e}_k$ respectively
     Let $H_{kk} = (y_+ + y_- - 2y)/r^2$         $\triangleright$ Diagonal entries
     **for** $\ell \leftarrow k+1$ to $d$ **do**
         Let $H_{k\ell} = H_{\ell k}$ be the average of $n$ samples of $(f(\boldsymbol{x} + r\boldsymbol{e}_k + r\boldsymbol{e}_\ell) + f(\boldsymbol{x} - r\boldsymbol{e}_k - r\boldsymbol{e}_\ell) - f(\boldsymbol{x} + r\boldsymbol{e}_k - r\boldsymbol{e}_\ell) - f(\boldsymbol{x} - r\boldsymbol{e}_k + r\boldsymbol{e}_\ell))/4r^2$         $\triangleright$ Off-diagonal entries
     **end for**
**end for**
Let $\hat{H}_0 = \{H_{jk}\}_{(i,j) \in [d]^2}$, and $\hat{H}$ be the matrix with same eigenvectors but with each eigenvalue $\lambda$ replaced by $\max\{\lambda, M\}$      $\triangleright$ Projecting to the set where $\hat{H} - MI_d$ is positive semidefinite
**Return** $\hat{H}$

---

**Theorem 4.1.** *For any fixed inputs $\boldsymbol{x}$, $Z$, $n$, and any function $f$ satisfying the Lipschitz Hessian condition with parameter $\rho$, the output $\hat{\boldsymbol{g}}$ returned by the GradientEst subroutine satisfies the following properties*

$$||\mathbb{E}[\hat{\boldsymbol{g}}] - Z\nabla f(\boldsymbol{x})||_2 \leq \frac{\lambda_Z^3 \rho \sqrt{d}}{2(d+2)}, \tag{2}$$

$$\mathrm{Tr}\left(\mathrm{Cov}[\hat{\boldsymbol{g}}]\right) \leq \frac{2d}{n}||Z\nabla f(\boldsymbol{x})||_2^2 + \frac{d^2}{18n}\left(\rho \lambda_Z^3\right)^2 + \frac{d^2}{2n}, \tag{3}$$

*where $\lambda_Z$ is the largest singular value of $Z$.*

**Remark 4.2.** *Inequality (2) provides a sharp characterization for the bias of the gradient estimator, as it can be matched for any $\lambda_Z$ and $d$ with a cubic polynomial $f$. Inequality (3) is sharp in the asymptotic regime when both $\nabla f$ and $\lambda_Z$ approaches zero.*

We also provide rough estimates on the high-probability bounds for the BootstrappingEst and the HessianEst functions. Specifically, we show that their errors have sub-Gaussain tails in distribution, as stated in the following theorem.

**Theorem 4.3.** *For any fixed inputs $\boldsymbol{x}$, $r$, $n$, any function $f$ satisfying the Lipschitz Hessian condition with parameter $\rho$, and any variable $K > 0$, the outputs $\hat{\boldsymbol{m}}$ and $\hat{H}$ returned by the BootstrappingEst and the HessianEst subroutine satisfy the following conditions.*

$$\mathbb{P}\left[\|\hat{\boldsymbol{m}} - \nabla f(\boldsymbol{x})\|_2 \geq K\right] \leq 2\exp\left(-\frac{K^2}{\frac{3d\rho^2 r^4}{4} + \frac{12d}{nr^2}}\right), \tag{4}$$

$$\mathbb{P}\left[\left\|\hat{H} - \nabla^2 f(\boldsymbol{x})\right\|_{\mathrm{F}} \geq K\right] \leq 2\exp\left(-\frac{K^2}{2d^2\rho^2 r^2 + \frac{144d^2}{nr^4}}\right). \tag{5}$$

We postpone the proof of the above theorems to Section 4.2 and Appendix C and proceed to describe how these results are used in the algorithm.

For brevity, let $\epsilon \triangleq \frac{\rho^{\frac{2}{3}}}{M} dT^{-\frac{2}{3}}$ be the minimax regret we aim to achieve, and let $\boldsymbol{x}_{\mathrm{B}}$ denote the estimator $\boldsymbol{x}$ stored at the end of the first stage. The role of the final stage is to ensure that if $f(\boldsymbol{x}_{\mathrm{B}}) - f(\boldsymbol{x}^*)$ is sufficiently small with high probability, the final result of the proposed algorithm achieves the stated simple regret guarantees. Formally, we require the following achievability result from the Bootstrapping stage.

**Theorem 4.4.** *For any fixed $\rho$, $M$ and $R$, the result returned by the first stage of Algorithm 4 satisfies*

$$\lim_{T \to \infty} \sup_{f \in \mathcal{F}(\rho, M, R)} \mathbb{E}\left[(f(\boldsymbol{x}_{\mathrm{B}}) - f(\boldsymbol{x}^*))^{\frac{3}{2}}\right] / \epsilon = 0. \tag{6}$$

Note that the above condition implies that $f(\boldsymbol{x}_{\mathrm{B}}) - f(\boldsymbol{x}^*)$ concentrates below $o\left(\epsilon^{\frac{2}{3}}\right)$, which is weaker than the $O(\epsilon)$ rates stated in our main theorems.[3] The bottleneck of the overall algorithm is on the final stage, and one can achieve equation (6) using any suboptimal algorithm with an expected simple regret of $o(T^{-\frac{4}{9}})$. For example, one can run the suboptimal algorithm twice, estimate their achieved function values by averaging over $o(T)$ samples, and then choose the outcome with the smaller estimated function value as $\boldsymbol{x}_{\mathrm{B}}$. In the rest of this section, we prove Theorem 3.1 assuming the correctness of the above theorem. A self-contained proof for Theorem 4.4 is provided in Appendix E.

Before proceeding with the proof, we provide a high-level description of the algorithm in the final stage. At the beginning, we perform a Hessian estimation near $\boldsymbol{x}_{\mathrm{B}}$ using the HessianEst subroutine with $O(T)$ samples. From Theorem 4.3, our choice of parameters results in an expected estimation error of $o(1)$ for sufficiently large $T$.

The algorithm proceeds to find a real matrix $Z_H$, which essentially serves as a linear transformation on the action domain such that the Hessian of the transformed function is approximately the identity matrix. Note that the projection step in the HessianEst function ensures the eigenvalues of the estimator are no less than $M$. There is always a valid solution of $Z_H$.

Then, we estimate the gradient at $\boldsymbol{x}_{\mathrm{B}}$ using the GradientEst subroutine, which samples on a hyperellipsoid with a shape characterized by $Z_H$. We chose the hyperellipsoid sampling in the final stage due to its superior performance in the small-gradient regime compared to coordinate-wise sampling. In contrast, the coordinate-wise estimator is used in the bootstrapping stage to eliminate the dependency of the local gradient on its bias-variance tradeoff, which is beneficial for the non-asymptotic analysis. Particularly, we scale the hyperellipsoid with a carefully designed factor (see the definition of variable $r_{\mathrm{g}}$) to minimize the estimation error. Then, the remaining steps can be interpreted as a modified Newton step, which essentially approximates the global minimum point with a quadratic approximation.

The analysis in our proof relies on the following proposition, which is proved in Appendix D.

**Proposition 4.5.** *For any given point $\boldsymbol{x}_{\mathrm{B}}$ and any function $f$ that satisfies strong convexity and Lipschitz Hessian, let $\tilde{\boldsymbol{x}} \triangleq \boldsymbol{x}_{\mathrm{B}} - (\nabla^2 f(\boldsymbol{x}_{\mathrm{B}}))^{-1}\nabla f(\boldsymbol{x}_{\mathrm{B}})$ and $\tilde{f}(\boldsymbol{x})$ denote the quadratic approximation*

---

[3]With more sophisticated analysis, this concentration requirement can be improved to only requiring a similar upper bound of $o\left(\epsilon^{\frac{1}{2}}\right)$. However, we choose equation (6) to provide a simpler proof, as it does not affect the asymptotic sample complexity.

**Algorithm 4** An Example Algorithm to Achieve the Minimax Rates

---

**Input** $T, \rho, M$
Let $\boldsymbol{x} = \boldsymbol{0}$
The First Stage:
**for** $k \leftarrow 1$ to $\lfloor T^{0.1} \rfloor$ **do**

    Let $n_{\mathrm{m}} = \lfloor \frac{T^{0.9}}{10d} \rfloor$, $n_{\mathrm{H}} = \lfloor \frac{T^{0.9}}{10d^2} \rfloor$, $r_{\mathrm{m}} = \left( \frac{8}{n_{\mathrm{m}}\rho^2} \right)^{\frac{1}{6}}$, $r_{\mathrm{H}} = \left( \frac{144}{n_{\mathrm{H}}\rho^2} \right)^{\frac{1}{6}}$

    Let $\hat{\boldsymbol{m}} = \mathrm{BootstrappingEst}(\boldsymbol{x}, r_{\mathrm{m}}, n_{\mathrm{m}})$, $\hat{H} = \mathrm{HessianEst}(\boldsymbol{x}, r_{\mathrm{H}}, n_{\mathrm{H}})$

    Let $H_{m^*}$ denote the matrix with the same eigenvectors of $\hat{H}$ but each eigenvalue $\lambda$ replaced by $\max\{\lambda, m^*\}$, choose $m^*$ to be the smallest value such that $||H_{m^*}^{-1}\hat{\boldsymbol{m}}||_2 \leq \frac{M}{\rho}$.

    Let $\boldsymbol{x} = \boldsymbol{x} - H_{m^*}^{-1}\hat{\boldsymbol{m}}$
**end for**
The Final Stage:

Let $n_{\mathrm{g}} = \lfloor \frac{T}{10} \rfloor$, $n_{\mathrm{H}} = \lfloor \frac{T}{10d^2} \rfloor$, $r_{\mathrm{g}} = \left( \frac{d^3}{n_{\mathrm{g}}\rho^2} \right)^{\frac{1}{6}}$, $r_{\mathrm{H}} = \left( \frac{144}{n_{\mathrm{H}}\rho^2} \right)^{\frac{1}{6}}$

Let $\hat{H} = \mathrm{HessianEst}(\boldsymbol{x}, r_{\mathrm{H}}, n_{\mathrm{H}})$, $Z_H$ be any symmetric matrix such that $Z_H^2 = \hat{H}^{-1}$, and $\lambda_{Z_H}$ be the largest eigenvalue of $Z_H$
Let $Z = r_{\mathrm{g}} Z_H / \lambda_{Z_H}$, $\hat{\boldsymbol{g}} = \mathrm{GradientEst}(\boldsymbol{x}, Z, n_{\mathrm{g}})$, $\boldsymbol{r} = -\hat{H}^{-1}Z^{-1}\hat{\boldsymbol{g}}$
Project $\boldsymbol{r}$ to the $L_2$ ball of radius $\frac{M}{\rho}$, i.e., $\boldsymbol{r} = \boldsymbol{r} \cdot \min\{1, \frac{M}{\rho ||\boldsymbol{r}||_2}\}$
**Return** $\boldsymbol{x} = \boldsymbol{x} + \boldsymbol{r}$

---

$\frac{1}{2}(\boldsymbol{x} - \tilde{\boldsymbol{x}})^{\intercal} \nabla^2 f(\boldsymbol{x}_{\mathrm{B}})(\boldsymbol{x} - \tilde{\boldsymbol{x}})$, *we have the following inequality for all $\boldsymbol{x}$ with $||\boldsymbol{x} - \boldsymbol{x}_{\mathrm{B}}||_2 \leq \frac{M}{\rho}$.*

$$f(\boldsymbol{x}) - f^* \leq 2\tilde{f}(\boldsymbol{x}) + \frac{12\rho(f(\boldsymbol{x}_{\mathrm{B}}) - f^*)^{\frac{3}{2}}}{M^{\frac{3}{2}}}. \tag{7}$$

*Furthermore, if $\boldsymbol{x}$ is generated by the final stage of Algorithm 4 with any parameter values that satisfy $n_{\mathrm{g}} \geq d^3$, $n_{\mathrm{H}} \geq \frac{64\rho^4 d^6}{M^6}$ and the first-stage output is set to $\boldsymbol{x}_{\mathrm{B}}$, then*

$$\mathbb{E}\left[ \tilde{f}(\boldsymbol{x}) \mid \boldsymbol{x}_{\mathrm{B}} \right] \leq \left( \frac{14d^2\rho^{\frac{4}{3}}}{M^2 n_{\mathrm{H}}^{\frac{1}{3}}} + \frac{52d}{n_{\mathrm{g}}} \right) (f(\boldsymbol{x}_{\mathrm{B}}) - f(\boldsymbol{x}^*)) + \frac{82\rho}{M^{\frac{3}{2}}}(f(\boldsymbol{x}_{\mathrm{B}}) - f(\boldsymbol{x}^*))^{\frac{3}{2}} + \frac{3d\rho^{\frac{2}{3}}}{M n_{\mathrm{g}}^{\frac{2}{3}}}. \tag{8}$$

Now, we use Proposition 4.5 to prove the achievability result.

*Proof of Theorem 3.1 given Theorem 4.4.* First, recall our construction ensures that $||\boldsymbol{x}_T - \boldsymbol{x}_{\mathrm{B}}||_2 \leq \frac{M}{\rho}$. Inequality (7) can always be applied and we have

$$\mathfrak{R}(T; \rho, M, R) \leq \sup_{f \in \mathcal{F}(\rho, M, R)} \mathbb{E}\left[ 2\tilde{f}(\boldsymbol{x}) + \frac{12\rho(f(\boldsymbol{x}_{\mathrm{B}}) - f^*)^{\frac{3}{2}}}{M^{\frac{3}{2}}} \right].$$

Then, when $T$ is sufficiently large, the conditions of (8) holds and we have

$$\mathfrak{R}(T; \rho, M, R) \leq \sup_{f \in \mathcal{F}(\rho, M, R)} \mathbb{E}\left[ \left( \frac{28d^2\rho^{\frac{4}{3}}}{M^2 n_{\mathrm{H}}^{\frac{1}{3}}} + \frac{104d}{n_{\mathrm{g}}} \right) (f(\boldsymbol{x}_{\mathrm{B}}) - f(\boldsymbol{x}^*)) + \frac{176\rho(f(\boldsymbol{x}_{\mathrm{B}}) - f^*)^{\frac{3}{2}}}{M^{\frac{3}{2}}} \right]$$

$$+ \frac{6d\rho^{\frac{2}{3}}}{M n_{\mathrm{g}}^{\frac{2}{3}}}.$$

Note that inequality (6) implies that $\mathbb{E}\left[ f(\boldsymbol{x}_{\mathrm{B}}) - f(\boldsymbol{x}^*) \right] = o(\epsilon^{\frac{2}{3}}) = o(T^{-\frac{4}{9}})$ and we have that $n_{\mathrm{H}}^{-\frac{1}{3}} + n_{\mathrm{g}}^{-1} = o(T^{-\frac{2}{9}})$. The RHS of the above inequality is dominated by the last term. Hence,

$$\limsup_{T \to \infty} \mathfrak{R}(T; \rho, M, R) \cdot T^{\frac{2}{3}} \leq \limsup_{T \to \infty} \frac{6d\rho^{\frac{2}{3}} T^{\frac{2}{3}}}{M n_{\mathrm{g}}^{\frac{2}{3}}} = O\left( \frac{\rho^{\frac{2}{3}}}{M} d \right).$$

To complete the proof, we show that the proposed algorithm samples the function values of $f$ at most $T - 1$ times. In the first stage, both BootstrappingEst and HessianEst are executed once per loop, with BootstrappingEst requiring at most $2dn_{\mathrm{m}} \leq \frac{T^{0.9}}{5}$ samples and HessianEst requiring at most $2d^2 n_{\mathrm{H}} \leq \frac{T^{0.9}}{5}$ samples each time. Therefore, the total number of samples used in the first stage is bounded by $\lfloor T \rfloor \left( \frac{T^{0.9}}{5} + \frac{T^{0.9}}{5} \right) \leq \frac{2T}{5}$. In the final stage, we make one call to both HessianEst and GradientEst, which together require $n_{\mathrm{H}}(2d^2) + 2n_{\mathrm{g}} \leq \frac{2T}{5}$ samples. Thus, the overall number of samples is bounded by $\frac{4T}{5}$, which ensures that it is no greater than $T - 1$. $\qquad \square$

### 4.2 Proof of Theorem 4.1

To prove inequality (2), we investigate the following function

$$\boldsymbol{G}(r; \boldsymbol{x}) \triangleq \mathbb{E}_{\boldsymbol{u} \sim \mathrm{Unif}(S^{d-1})} \left[ \frac{d}{2r} (f(\boldsymbol{x} + r\boldsymbol{u}) - f(\boldsymbol{x} - r\boldsymbol{u})) \boldsymbol{u} \right],$$

where $\mathrm{Unif}(S^{d-1})$ denotes the uniform distribution on $S^{d-1}$. Recall that in our algorithm we have $\mathbb{E}[\hat{\boldsymbol{g}}] = r\boldsymbol{G}(r; \boldsymbol{x})$ if $Z = rI_d$ for some $r \in (0, +\infty)$, and by differentiability we have $\nabla f(\boldsymbol{x}) = \lim_{z \to 0^+} \boldsymbol{G}(z; \boldsymbol{x})$. Under this condition, we can bound $\|\mathbb{E}[\hat{\boldsymbol{g}}] - r\nabla f(\boldsymbol{x})\|_2$ by integration, i.e.,

$$\|\mathbb{E}[\hat{\boldsymbol{g}}] - r\nabla f(\boldsymbol{x})\|_2 = r \left\| \boldsymbol{G}(r; \boldsymbol{x}) - \lim_{z \to 0^+} \boldsymbol{G}(z; \boldsymbol{x}) \right\|_2 \leq r \int_{0^+}^r \left\| \frac{d}{dz} \boldsymbol{G}(z; \boldsymbol{x}) \right\|_2 dz. \quad (9)$$

Note that $\boldsymbol{G}(z; \boldsymbol{x})$ can be written into the following equivalent form.

$$\boldsymbol{G}(z; \boldsymbol{x}) = \frac{\int_{S^{d-1}} \frac{d}{2z} (f(\boldsymbol{x} + z\boldsymbol{u}) - f(\boldsymbol{x} - z\boldsymbol{u})) \mathbf{dA}}{\int_{S^{d-1}} \|\mathbf{dA}\|_2},$$

where the integration is with respect to $\boldsymbol{u}$ over the surface $S^{d-1}$, and $\mathbf{dA}$ is the vector surface element, i.e., with the magnitude being the infinitesimally small surface area and the direction perpendicular to the surface (pointing outward). The differential of $\boldsymbol{G}(z; \boldsymbol{x})$ over $z$ can be written as

$$\begin{aligned}
\frac{d}{dz} \boldsymbol{G}(z; \boldsymbol{x}) &= \frac{\int_{S^{d-1}} \frac{\partial}{\partial z} \left( \frac{d}{2z} (f(\boldsymbol{x} + z\boldsymbol{u}) - f(\boldsymbol{x} - z\boldsymbol{u})) \right) \mathbf{dA}}{\int_{S^{d-1}} \|\mathbf{dA}\|_2} \\
&= \frac{\int_{S^{d-1}} -\frac{d}{2z^2} (f(\boldsymbol{x} + z\boldsymbol{u}) - f(\boldsymbol{x} - z\boldsymbol{u})) \mathbf{dA}}{\int_{S^{d-1}} \|\mathbf{dA}\|_2} \\
&\quad + \frac{\int_{S^{d-1}} \frac{d}{2z} \boldsymbol{u} \cdot (\nabla f(\boldsymbol{x} + z\boldsymbol{u}) + \nabla f(\boldsymbol{x} - z\boldsymbol{u})) \mathbf{dA}}{\int_{S^{d-1}} \|\mathbf{dA}\|_2}.
\end{aligned}$$

The gist of this proof is to note that for any $\boldsymbol{u} \in S$ we have $\boldsymbol{u}$ and $\mathbf{dA}$ are parallel (i.e., $\boldsymbol{u}$ is parallel to the normal vector of the hypersphere at the same point), so the second term in the integral above on the numerator can be written as

$$\int_{S^{d-1}} \frac{d}{2z} \boldsymbol{u}(\nabla f(\boldsymbol{x} + z\boldsymbol{u}) + \nabla f(\boldsymbol{x} - z\boldsymbol{u})) \cdot \mathbf{dA}.$$

Hence, by divergence theorem, we have

$$\begin{aligned}
\frac{d}{dz} \boldsymbol{G}(z; \boldsymbol{x}) &= \frac{1}{\int_{S^{d-1}} \|\mathbf{dA}\|_2} \cdot \left( \int_{B^d} \nabla_{\boldsymbol{u}} \cdot \left( -\frac{d}{2z^2} I_d \left( f(\boldsymbol{x} + z\boldsymbol{u}) - f(\boldsymbol{x} - z\boldsymbol{u}) \right) \right. \right. \\
&\quad \left. \left. + (\nabla f(\boldsymbol{x} + z\boldsymbol{u}) + \nabla f(\boldsymbol{x} - z\boldsymbol{u})) \frac{d}{2z} \boldsymbol{u} \right) \mathbf{dV} \right) \\
&= \frac{d}{2} \cdot \frac{\int_{B^d} \boldsymbol{u} \operatorname{Tr}(\nabla^2 f(\boldsymbol{x} + z\boldsymbol{u}) - \nabla^2 f(\boldsymbol{x} - z\boldsymbol{u})) \mathbf{dV}}{\int_{S^{d-1}} \|\mathbf{dA}\|_2}, \quad (10)
\end{aligned}$$

where $B^d$ denotes the standard hyperball.

Now consider any unit vector $e$. Let $u_e$ denote the reflection of $u$ with respect to the hyperplane orthogonal to $e$, i.e., $u_e \triangleq u - 2(u \cdot e)e$. Because the hyperball $B$ is invariant under the reflection $u \to u_e$, equation (10) can also be written as

$$\frac{d}{dz}G(z; x) = \frac{d}{2} \cdot \frac{\int_{B^d} u_e \operatorname{Tr}(\nabla^2 f(x + zu_e) - \nabla^2 f(x - zu_e))dV}{\int_{S^{d-1}} ||dA||_2}. \tag{11}$$

Hence, by averaging equation (10) and (11), we have

$$\begin{aligned}
\frac{d}{dz}G(z; x) \cdot e &= \frac{d}{4} \frac{\int_{B^d} u \operatorname{Tr}(\nabla^2 f(x + zu) - \nabla^2 f(x - zu))dV}{\int_{S^{d-1}} ||dA||_2} \cdot e \\
&\quad + \frac{d}{4} \frac{\int_{B^d} u_e \operatorname{Tr}(\nabla^2 f(x + zu_e) - \nabla^2 f(x - zu_e))dV}{\int_{S^{d-1}} ||dA||_2} \cdot e \\
&= \frac{d}{4} \frac{\int_{B^d} u \cdot e \operatorname{Tr}(\nabla^2 f(x + zu) - \nabla^2 f(x + zu_e))dV}{\int_{S^{d-1}} ||dA||_2} \\
&\quad + \frac{d}{4} \frac{\int_{B^d} -u \cdot e \operatorname{Tr}(\nabla^2 f(x - zu) - \nabla^2 f(x - zu_e))dV}{\int_{S^{d-1}} ||dA||_2}. \tag{12}
\end{aligned}$$

By the Lipschitz Hessian condition and Cauchy's inequality, the difference between the differential terms above can be bounded as follows.

$$\begin{aligned}
\left| \operatorname{Tr}\left(\nabla^2 f(x \pm zu) - \nabla^2 f(x \pm zu_e)\right) \right| &\leq \sqrt{d} ||\nabla^2 f(x \pm zu) - \nabla^2 f(x \pm zu_e)||_F \\
&\leq \rho\sqrt{d} ||zu - zu_e||_2 = 2z\rho\sqrt{d}|u \cdot e|. \tag{13}
\end{aligned}$$

Consequently,

$$\left| \frac{d}{dz}G(z; x) \cdot e \right| \leq \frac{z\rho d\sqrt{d} \int_{B^d} (u \cdot e)^2 dV}{\int_{S^{d-1}} ||dA||_2} = \frac{z\rho\sqrt{d}}{d+2}.$$

Note that $e$ can be any unit vector. We have essentially bounded the $\ell_2$ norm of $\frac{d}{dz}G(z; x)$, i.e.,

$$\left\| \frac{d}{dz}G(z; x) \right\|_2 \leq \frac{z\rho\sqrt{d}}{d+2}.$$

As mentioned earlier, when $Z = rI_d$ inequality (2) is obtained by applying this gradient-norm bound to inequality (9).

For general input matrix $Z$, we can view GradientEst as a subroutine that operates on the same function $f$ but with a linear transformation applied to the input domain. Formally, let $f'(y) \triangleq f(x + \frac{Z}{\lambda_Z}(y - x))$. We have that $f'$ satisfies the Lipschitz Hessian condition with parameter $\rho$ as well. Therefore, inequality (2) can be obtained following the same analysis by replacing $f$ with $f'$ and $Z$ with $\lambda_Z I_d$.

Now we present the proof for inequality (3). Formally, let $w_+$, $w_-$ be two independent samples of additive noises. Then the trace of covariance matrix of $\hat{g}$ can upper bounded using the second moments of single measurements.

$$\begin{aligned}
\operatorname{Tr}(\operatorname{Cov}[\hat{g}]) &\leq \frac{1}{n} \mathbb{E}_{u \sim \operatorname{Unif}(S^{d-1}), w_+, w_-} \left[ \left(\frac{d}{2}\right)^2 \left(f(x + Zu) - f(x - Zu) + w_- - w_-\right)^2 \right] \\
&= \frac{d^2}{4n} \mathbb{E}_{u \sim \operatorname{Unif}(S^{d-1})} \left[ \left(f(x + Zu) - f(x - Zu)\right)^2 + 2 \right]. \tag{14}
\end{aligned}$$

The identity above uses the fact that additive noises are unbiased and have bounded variances.

Note that from the Lipschitz Hessian condition, we have that

$$|f(x \pm Zu) - f_2(x \pm Zu)| \leq \frac{1}{6}\rho ||Zu||_2^3 \leq \frac{1}{6}\rho\lambda_Z^3,$$

where $f_2$ is the Taylor polynomial of $f$ expanded at $\boldsymbol{x}$ up to the quadratic terms. Consequently, inequality (14) implies

$$
\begin{aligned}
\mathrm{Tr}\left(\mathrm{Cov}[\hat{\boldsymbol{g}}]\right) &\leq \frac{d^2}{4n}\mathbb{E}\left[\left(|f_2(\boldsymbol{x}+Z\boldsymbol{u})-f_2(\boldsymbol{x}-Z\boldsymbol{u})|+\frac{1}{3}\rho\lambda_Z^3\right)^2+2\right] \\
&= \frac{d^2}{4n}\mathbb{E}\left[\left(|2Z\boldsymbol{u}\cdot\nabla f(\boldsymbol{x})|+\frac{1}{3}\rho\lambda_Z^3\right)^2+2\right] \\
&\leq \frac{d^2}{4n}\mathbb{E}\left[2\cdot|2Z\boldsymbol{u}\cdot\nabla f(\boldsymbol{x})|^2+2\left(\frac{1}{3}\rho\lambda_Z^3\right)^2+2\right] \\
&= \frac{2d}{n}\|Z\nabla f(\boldsymbol{x})\|_2^2+\frac{d^2}{18n}\left(\rho\lambda_Z^3\right)^2+\frac{d^2}{2n}
\end{aligned}
$$

where the expectations are taken of $\boldsymbol{u}\sim\mathrm{Unif}(S^{d-1})$, and the last equality is due to the well-known fact that $\mathbb{E}\left[\boldsymbol{u}\boldsymbol{u}^{\mathsf{T}}\right]=\frac{1}{d}I_d$.

## 5    Conclusion and Future Work

In this work, we achieve the first minimax simple regret for bandit optimization of second-order smooth and strongly convex functions. We derived the matching upper and lower bounds and proposed an algorithm that integrates a bootstrapping stage with a mirror-descent stage. Our key technical innovations include a sharp characterization of the spherical-sampling gradient estimator under higher-order smoothness conditions and a novel iterative method for the bootstrapping stage that remains effective with unbounded Hessians.

While these advancements settle the fundamental problem of optimizing second-order smooth and strongly convex functions with zeroth-order feedback, the techniques and insights presented in this paper also pave the way for further research in this domain. One interesting follow-up direction is to generalize our analysis to the online setting for the average regret metric. Additionally, investigating the fundamental tradeoff between simple regret and average regret could yield valuable insights for task-specific algorithmic designs.

## Acknowledgement

JDL acknowledges the support of the NSF CCF 2002272, NSF IIS 2107304, and NSF CAREER Award 2144994.

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

## A    Detailed Comparison on Different Smoothness Conditions

In a relevant line of work, the higher-order smoothness of the objective function in the $k = 3$ case is characterized by the following generalized Hölder condition.

$$\left| f(\boldsymbol{z}) - f(\boldsymbol{x}) - \nabla f(\boldsymbol{z})(\boldsymbol{z} - \boldsymbol{x}) - \frac{1}{2}(\boldsymbol{z} - \boldsymbol{x})^\intercal \left( \nabla^2 f(\boldsymbol{z}) \right) (\boldsymbol{z} - \boldsymbol{x}) \right| \le L \|\boldsymbol{z} - \boldsymbol{x}\|_2^3.$$

Note that the $\rho$-Lipschitz Hessian condition in our work implies an Hölder condition with parameter $L = \frac{\rho}{6}$. A direct application of the works in Table 1 requires order-wise larger sample complexities in our setting on top of the additional $k = 2$ smoothness condition. On the other hand, the $L$-Holder condition implies $\rho = O(\sqrt{d}L)$-Lipschitz Hessian. Hence, a direct application of our algorithm order-wise improves the sample complexity in the setting of generalized Hölder condition in a polynomial factor of $d$ as well.

In terms of the characterization of gradient estimators, the prior works of Bach & Perchet (2016); Akhavan et al. (2020); Novitskii & Gasnikov (2021) used isotropic sampling over a bounded set of radius $r$ and presented upper bounds on the estimation bias of $O(Lr^2)$, $O(Ldr^2)$, and $O(L\sqrt{d}r^2)$, respectively. In this special case, our Theorem 4.1 implies an upper bound of $O(\rho r^2 / \sqrt{d})$, and similar to the above analysis, this bound strengths the bounds in prior works.

## B    Proof of Theorem 3.2

To illustrate the proof idea, we start with the case of $d = 1$.

### B.1    Illustrating example: 1D case

The gist of our proof is to construct a pair of hard-instance functions that need to sufficiently distant from each other to avoid trivial optimizers with low simple regret. We also require them to be sufficiently close to each other so that they are indistinguishable without sufficiently many samples. These requirements are captured quantitatively in the following result, which is proved using an analysis of KL divergence. Here we assume their correctness and focus on the construction.

**Definition B.1.** For any (Borel measurable) function class $\mathcal{F}_\mathrm{H}$ and any distribution $p$ defined on $\mathcal{F}_\mathrm{H}$, we define the *uniform sampling error* to be

$$P_\epsilon \triangleq \inf_{\boldsymbol{x}} \mathbb{P}_{f \sim p}[f(\boldsymbol{x}) - \inf f \ge \epsilon].$$

We also define the *maximum local variance* to be

$$V \triangleq \sup_{\boldsymbol{x}} \operatorname{Var}_{f \sim p}[f(x)].$$

**Lemma B.2** (Restatement of Proposition 7 in Yu et al. (2023b)). *For any sampling algorithm to achieve an expected simple regret of $\epsilon > 0$ over a function class $\mathcal{F}_\mathrm{H}$, if $P_{2\epsilon/c} \ge c$ for some universal constant $c \in (0, 1)$, and the observation noises are standard Gaussian, then the required sample complexity to achieve a minimax regret of $\epsilon$ is at least $\Omega(1/V)$.*

We construct our hard instances using the following function

$$g(x) = \begin{cases} \frac{1}{2} \left( \sin\left(\frac{1}{2}x\right) + 1 \right) & \text{if } x \in (-\pi, 3\pi] \\ -\cos x - 1 & \text{if } x \in (-3\pi, -\pi] \\ 0 & \text{otherwise.} \end{cases}$$

Some key properties of $g(x)$ to be used are that its differential $g'(x)$ is 1-Lipschitz, and we have $|g'(x)| \le 1$ for all $x$. Our hard instances consist of two functions. We define

$$f_1(x) = Mx^2 + y_0 \int_{-\pi}^{x/x_0} g(z)dz,$$

$$f_2(x) = Mx^2 + y_0 \int_{-\pi}^{-x/x_0} g(z)dz,$$

where $y_0$, $x_0$ are normalization factors given by $y_0 = \frac{1}{\pi\sqrt{T}}$, $x_0 = \left(\frac{y_0}{\rho}\right)^{\frac{1}{3}}$. The normalization factors are chosen to satisfy the Lipschitz Hessian condition and a maximum local variance bound required for a KL-divergence based approach presented in Lemma B.2.

Specifically, the choice of $x_0$ and the fact that $g'(x)$ is 1-Lipschitz imply that both $f_1$ and $f_2$ satisfy the Lipschitz Hessian condition. Then because the absolute value of integration of $g(x)$ is bounded by $2\pi$, one can show that the maximum local variance for the function class $\{f_1, f_2\}$ is no greater than $\pi^2 y_0^2 = \frac{1}{T}$ for the uniform prior distribution, which is to be used to show the sample complexity lower bound.

We first check that both $f_1$ and $f_2$ are within our function class of interests. Note that both $f_1''(x)$ and $f_2''(x)$ belong to the interval $\left[2M - \frac{5}{4}\frac{y_0}{x_0^2}, 2M - \frac{3}{4}\frac{y_0}{x_0^2}\right]$. From the fact that $\lim_{T\to\infty}\frac{y_0}{x_0^2} = 0$ and $M > 0$, we have both $f_1''(x) > M$ and $f_2''(x) > M$ for all $x$ for sufficiently large $T$. So the strong convexity requirement is satisfied. On the other hand, consider any global minimum point $x^*$ of either $f_1$ or $f_2$. Because of their differentiability, we must have $f_1'(x) = 0$ or $f_2'(x) = 0$. Note that for all $x$, we have $|g(x)| \leq 2$, and

$$f_1'(x) = 2Mx + g\left(\frac{x}{x_0}\right)\frac{y_0}{x_0}$$
$$f_2'(x) = 2Mx - g\left(\frac{x}{x_0}\right)\frac{y_0}{x_0}.$$

We must have $|x^*| \leq \frac{y_0}{x_0}/M$, where the RHS is $o(1)$ for large $T$. Combined with strong convexity, this inequality implies that assumption A3 holds for both functions. To conclude, we have proved that $f_1, f_2 \in \mathcal{F}(\rho, M, R)$ for sufficiently large $T$.

Now we let $\epsilon = \frac{1}{128M}\left(\frac{y_0}{x_0}\right)^2$ and $c = \frac{1}{2}$ to apply Lemma B.2. Note that

$$\liminf_{T\to\infty} T^{\frac{2}{3}}\epsilon = \frac{\rho^{\frac{2}{3}}}{128\pi^{\frac{4}{3}}M}.$$

The quantity $\epsilon$ exactly matches the lower bounds we aim to prove. Therefore, it remains to check that the required condition on uniform sampling errors in Definition B.1 are satisfied.

Formally, we need to show that $f_k(0) - \inf_x f_k(x) \geq 4\epsilon$ for $k \in \{1, 2\}$, so that the uniform sampling error $P_{4\epsilon}$ under uniform distribution over $\mathcal{F}_H$ is lower bounded by $\frac{1}{2}$ and Lemma B.2 can be applied. Without loss of generality, we focus on the case of $k = 1$. Note that $f_1''(x) \leq 2M + \frac{y_0}{4x_0^2}$ for all $x \in [-\pi x_0, 0]$. Therefore, we have

$$f_1(x) - f_1(0) \leq f_1'(0)x + \frac{1}{2}x^2 \sup_{z\in[-\pi x_0, 0]} f_1''(z)$$
$$\leq \frac{y_0}{2x_0}x + \frac{1}{2}x^2\left(2M + \frac{y_0}{4x_0^2}\right)$$

for $x \in [-\pi x_0, 0]$, and $\lim_{T\to\infty} x_0 = 0$. Consider any sufficiently large $T$ such that $\frac{y_0}{4x_0^2} \leq 2M$, we can choose $x = -\frac{y_0}{2x_0}\frac{1}{2M + \frac{y_0}{4x_0^2}}$ for the above bound, which falls into the interval of $[-\pi x_0, 0]$. Then we have

$$\inf_x f_1(x) \leq f_1\left(-\frac{y_0}{2x_0}\frac{1}{2M + \frac{y_0}{4x_0^2}}\right)$$
$$\leq f_1(0) - \frac{1}{2}\left(\frac{y_0}{2x_0}\right)^2\frac{1}{2M + \frac{y_0}{4x_0^2}}$$
$$\leq f_1(0) - 4\epsilon.$$

We use this inequality to lower bound the minimum sampling error. Note that $f_1$ is an increasing function for $x \geq 0$ and $\inf_x f_1(x) = \inf_x f_2(x)$. We have $f_1(x) \geq \inf_x f_2(x) + 4\epsilon$ for $x \geq 0$.

Following the same arguments, we also have $f_2(x) \geq \inf_x f_1(x) + 4\epsilon$ for $x \leq 0$. Recall the definition of uniform sampling error in Definition B.1. We have essentially proved that $P_{4\epsilon} \geq \frac{1}{2}$. According to earlier discussions, this implies that the minimax simple regret is lower bounded by

$$\epsilon = \Omega\left(\frac{\rho^{\frac{2}{3}} T^{-\frac{2}{3}}}{M}\right).$$

### B.2 Proof for the General Case

The generalization of the earlier 1D lower bound is obtained by constructing a set of hard-instance functions where the optimization problem over this subset consists of $d$ binary hypothesis estimation problems, each identical to a 1D construction. Formally, for any $\boldsymbol{s} = (s_1, s_2, ..., s_d) \in \{1, 2\}^d$ and any input $\boldsymbol{x} = (x_{(1)}, x_{(2)}, ..., x_{(d)})$, we let

$$f_{\boldsymbol{s}}(\boldsymbol{x}) = \sum_{j=1}^d f_{s_j}(x_{(j)}).$$

One can verify that $f_{\boldsymbol{s}} \in \mathcal{F}(\rho, M, R)$ for all $\boldsymbol{s}$ for sufficiently large $T$.

Note that the simple regret for the above function class can be written as the sum of $d$ individual terms $\sum_{j=1}^d \left(f_{s_j}(x_{(j)}) - \inf_x f_{s_j}(x)\right)$. As proved earlier, the expectation of each term associated with any index $j$ is at least $\Omega\left(\frac{\rho^{\frac{2}{3}} T^{-\frac{2}{3}}}{M}\right)$ even if all entries of $\boldsymbol{s}$ except $s_j$ is known. Therefore, the total expected regret is lower bounded by $\Omega\left(\frac{d\rho^{\frac{2}{3}} T^{-\frac{2}{3}}}{M}\right)$.

## C  Proof of Theorem 4.3

We use the following elementary facts, which are versions of well-known properties of subgaussian and subexponential distributions in Vershynin (2018), but with explicit and possibly improved constant factors. For completeness, we provide their proofs in Appendix F.

**Proposition C.1.** *For any real-valued zero-mean independent random variables $z_1, ..., z_k$, if*

$$\mathbb{P}[|z_j| \geq K] \leq 2\exp\left(-\frac{K^2}{\sigma_j^2}\right) \qquad \forall j \in [k], K \in [0, +\infty), \tag{15}$$

*for some $\sigma_1, ..., \sigma_k$, then*

$$\mathbb{P}\left[\left|\sum_{j=1}^k z_j\right| \geq K\right] \leq 2\exp\left(-\frac{K^2}{4\sum_{j=1}^k \sigma_j^2}\right) \qquad \forall K \in [0, +\infty). \tag{16}$$

**Proposition C.2.** *For any real-valued independent random variables $z_1, ..., z_k$, if*

$$\mathbb{P}[|z_j| \geq K] \leq 2\exp\left(-\frac{K}{\sigma_j}\right) \qquad \forall j \in [k], K \in [0, +\infty), \tag{17}$$

*for some positive $\sigma_1, ..., \sigma_k$, then*

$$\mathbb{P}\left[\left|\sum_{j=1}^k z_j\right| \geq K\right] \leq 2\exp\left(-\frac{K}{3\sum_{j=1}^k \sigma_j}\right) \qquad \forall K \in [0, +\infty). \tag{18}$$

*Proof of equation (4).* We first prove the bound entry-vise. Consider any $m_k$, which contains a summation of $2n$ independent subgaussian variables. By Prop. C.1 we have that

$$\mathbb{P}\left[|m_k - \mathbb{E}[m_k]| \geq K\right] \leq 2\exp\left(-\frac{K^2}{2/nr^2}\right) \qquad \forall K \in [0, +\infty). \tag{19}$$

Then, by the Lipschitz Hessian condition, the bias for each entry is bounded as follows.

$$\left| \mathbb{E}\left[ m_k \right] - \frac{\partial}{\partial x_k} f(\boldsymbol{x}) \right| \leq \frac{1}{6} \rho r^2, \tag{20}$$

where $x_k$ denotes the $k$th entry of $\boldsymbol{x}$. Hence, each $\left| m_k - \frac{\partial}{\partial x_k} f(\boldsymbol{x}) \right|^2$ is subexpoential, i.e.,

$$\mathbb{P}\left[ \left| m_k - \frac{\partial}{\partial x_k} f(\boldsymbol{x}) \right|^2 \geq K^2 \right] \leq \mathbb{P}\left[ |m_k - \mathbb{E}\left[ m_k \right]| \geq K - \left| \mathbb{E}\left[ m_k \right] - \frac{\partial}{\partial x_k} f(\boldsymbol{x}) \right| \right]$$

$$\leq \mathbb{P}\left[ |m_k - \mathbb{E}\left[ m_k \right]| \geq K - \frac{1}{6} \rho r^2 \right]$$

$$\leq \max\left\{ 2 \exp\left( -\frac{\left( \max\{K - \frac{1}{6}\rho r^2, 0\} \right)^2}{2/nr^2} \right), 1 \right\}$$

$$\leq 2 \exp\left( -\frac{K^2}{\frac{\rho^2 r^4}{4} + \frac{4}{nr^2}} \right). \tag{21}$$

By the independence of $m_k$'s, we can apply Prop. C.2 to the inequality. Therefore,

$$\mathbb{P}\left[ \|\hat{\boldsymbol{m}} - \nabla f(\boldsymbol{x})\|_2 \geq K \right] = \mathbb{P}\left[ \left| \sum_{k=1}^{d} |m_k - \frac{\partial}{\partial x_k} f(\boldsymbol{x})|^2 \right| \geq K^2 \right]$$

$$\leq 2 \exp\left( -\frac{K^2}{\frac{3d\rho^2 r^4}{4} + \frac{12d}{nr^2}} \right) \qquad \forall K \in [0, +\infty). \tag{22}$$

$\square$

*Proof of equation* (5). We first provide the entry-wise bounds for the intermediate estimator $\hat{H}_0$. Each diagonal entry $H_{kk}$ contains the weighted average of $3n$ subgaussian variables. Conditioned on any realization of $y$, which is shared among all diagonal elements, Prop. C.1 can be applied for the rest of the $2n$ terms, and provides the following bounds.

$$\mathbb{P}[|H_{kk} - \mathbb{E}[H_{kk}|y]| \geq K] \leq 2 \exp\left( -\frac{K^2}{8/nr^4} \right) \qquad \forall K \in [0, +\infty). \tag{23}$$

Then, because the off-diagonal entries are independent, we have the following bounds for any $j \neq k$.

$$\mathbb{P}[|H_{jk} - \mathbb{E}[H_{jk}]| \geq K] \leq 2 \exp\left( -\frac{K^2}{1/nr^4} \right) \qquad \forall K \in [0, +\infty). \tag{24}$$

Hence, similar to the earlier proof steps, Prop. C.2 implies that

$$\mathbb{P}\left[ \|\hat{H}_0 - \mathbb{E}[\hat{H}_0|y]\|_{\mathrm{F}}^2 \geq K^2 \right] \leq 2 \exp\left( -\frac{K^2}{6d(d+3)/nr^4} \right)$$

$$\leq 2 \exp\left( -\frac{K^2}{24d^2/nr^4} \right). \tag{25}$$

Now, we take into account the estimation bias and the error of $y$. By Lipschitz Hessian, it is clear that

$$\left| H_{jk} - \frac{\partial}{\partial x_j} \frac{\partial}{\partial x_k} f(\boldsymbol{x}) \right| \leq \begin{cases} \frac{1}{3} \rho r & \text{if } j = k, \\ \frac{\sqrt{2}}{3} \rho r & \text{otherwise.} \end{cases}$$

Hence,

$$\left\| \hat{H}_0 - \nabla^2 f(\boldsymbol{x}) \right\|_{\mathrm{F}} \leq \frac{\sqrt{2} d \rho r}{3}. \tag{26}$$

Furthermore, note that $\mathbb{E}[\hat{H}_0] - \mathbb{E}[\hat{H}_0|y] = 2(y - f(\boldsymbol{x}))I_d/r^2$, where $I_d$ denotes the identity matrix. The subgaussian condition and Prop. C.1 imply that

$$\mathbb{P}[||\mathbb{E}[\hat{H}_0] - \mathbb{E}[\hat{H}_0|y]||_{\mathrm{F}} \geq K] = \mathbb{P}\left[|y - f(\boldsymbol{x})| \geq \frac{Kr^2}{2\sqrt{d}}\right]$$

$$\leq 2\exp\left(-\frac{K^2}{8d/nr^4}\right) \qquad \forall\, K \in [0, +\infty). \qquad (27)$$

We can combine the above bounds using triangle inequality and the union bound. Specifically, from inequalities (25), (26), and (27), we have the following bound for any $K \geq \frac{\sqrt{2}d\rho r}{3}$.

$$\mathbb{P}\left[||\hat{H}_0 - \nabla^2 f(\boldsymbol{x})||_{\mathrm{F}} \geq K\right] \leq \mathbb{P}\left[||\hat{H}_0 - \mathbb{E}[\hat{H}_0]||_{\mathrm{F}} \geq \frac{2}{3}\left(K - \frac{\sqrt{2}d\rho r}{3}\right)\right]$$

$$\leq \mathbb{P}\left[||\hat{H}_0 - \mathbb{E}[\hat{H}_0|y]||_{\mathrm{F}} \geq \frac{1}{3}\left(K - \frac{\sqrt{2}d\rho r}{3}\right)\right]$$

$$+ \mathbb{P}\left[||\mathbb{E}[\hat{H}_0] - \mathbb{E}[\hat{H}_0|y]||_{\mathrm{F}} \geq \left(K - \frac{\sqrt{2}d\rho r}{3}\right)\right]$$

$$\leq 2\exp\left(-\frac{\left(K - \frac{\sqrt{2}d\rho r}{3}\right)^2}{54d^2/nr^4}\right) + 2\exp\left(-\frac{\left(K - \frac{\sqrt{2}d\rho r}{3}\right)^2}{72d/nr^4}\right).$$

Utilize the fact that any probability measure is no greater than 1, the above inequality implies that

$$\mathbb{P}\left[||\hat{H}_0 - \nabla^2 f(\boldsymbol{x})||_{\mathrm{F}} \geq K\right] \leq 2\exp\left(-\frac{\left(\max\left\{K - \frac{\sqrt{2}d\rho r}{3}, 0\right\}\right)^2}{128d^2/nr^4}\right)$$

$$\leq 2\exp\left(-\frac{K^2}{2d^2\rho^2 r^2 + \frac{144d^2}{nr^4}}\right) \qquad \forall K \in [0, +\infty). \quad (28)$$

Finally, the needed bound for $\hat{H}$ is due to the projection to a convex set where the target $\nabla^2 f(\boldsymbol{x})$ belongs. Hence, the distance is not increased w.p.1, i.e., we always have $||\hat{H} - \nabla^2 f(\boldsymbol{x})||_{\mathrm{F}} \leq ||\hat{H}_0 - \nabla^2 f(\boldsymbol{x})||_{\mathrm{F}}$. □

## D Proof of Proposition 4.5

Inequality (7) is derived from the following approximations, which are due to the Lipschitz Hessian condition at $\boldsymbol{x}_{\mathrm{B}}$.

$$f(\boldsymbol{x}) \leq f(\boldsymbol{x}_{\mathrm{B}}) + \tilde{f}(\boldsymbol{x}) - \tilde{f}(\boldsymbol{x}_{\mathrm{B}}) + \frac{1}{6}\rho||\boldsymbol{x} - \boldsymbol{x}_{\mathrm{B}}||_2^3, \qquad (29)$$

$$f(\boldsymbol{x}^*) \geq f(\boldsymbol{x}_{\mathrm{B}}) + \tilde{f}(\boldsymbol{x}^*) - \tilde{f}(\boldsymbol{x}_{\mathrm{B}}) - \frac{1}{6}\rho||\boldsymbol{x}^* - \boldsymbol{x}_{\mathrm{B}}||_2^3. \qquad (30)$$

Noting that $\tilde{f}(\boldsymbol{x}^*) \geq 0$, the above inequalities imply that

$$f(\boldsymbol{x}) - f(\boldsymbol{x}^*) \leq \tilde{f}(\boldsymbol{x}) + \frac{1}{6}\rho||\boldsymbol{x} - \boldsymbol{x}_{\mathrm{B}}||_2^3 + \frac{1}{6}\rho||\boldsymbol{x}^* - \boldsymbol{x}_{\mathrm{B}}||_2^3. \qquad (31)$$

By strong convexity, we have $||\boldsymbol{x}^* - \boldsymbol{x}_{\mathrm{B}}||_2^2 \leq \frac{2(f(\boldsymbol{x}_{\mathrm{B}}) - f(\boldsymbol{x}^*))}{M}$. Hence, it remains to provide an upper bound for $\frac{1}{6}\rho||\boldsymbol{x} - \boldsymbol{x}_{\mathrm{B}}||_2^3$.

When $||\boldsymbol{x} - \boldsymbol{x}_{\mathrm{B}}||_2 \leq \sqrt{3}||\boldsymbol{x} - \tilde{\boldsymbol{x}}||_2$, we apply the condition $||\boldsymbol{x} - \boldsymbol{x}_{\mathrm{B}}||_2 \leq \frac{M}{\rho}$ to obtain that

$$\frac{1}{6}\rho||\boldsymbol{x} - \boldsymbol{x}_{\mathrm{B}}||_2^3 \leq \frac{1}{2}M||\boldsymbol{x} - \tilde{\boldsymbol{x}}||_2^2 \leq \tilde{f}(\boldsymbol{x}),$$

where the last step is due to the strong convexity of $\tilde{f}$. This implies inequality (7).

For the other case, we have $||\boldsymbol{x} - \boldsymbol{x}_{\mathrm{B}}||_2 \geq \sqrt{3}||\boldsymbol{x} - \tilde{\boldsymbol{x}}||_2$. We replace the variable $\boldsymbol{x}$ in inequality (29) with $\tilde{\boldsymbol{x}}$ to obtain that

$$f(\boldsymbol{x}^*) \leq f(\tilde{\boldsymbol{x}}) \leq f(\boldsymbol{x}_{\mathrm{B}}) - \tilde{f}(\boldsymbol{x}_{\mathrm{B}}) + \frac{1}{6}\rho||\tilde{\boldsymbol{x}} - \boldsymbol{x}_{\mathrm{B}}||_2^3. \tag{32}$$

By strong convexity,

$$\tilde{f}(\boldsymbol{x}_{\mathrm{B}}) \geq \frac{1}{2}M||\tilde{\boldsymbol{x}} - \boldsymbol{x}_{\mathrm{B}}||_2^2, \tag{33}$$

and by triangle inequality, we have that

$$||\tilde{\boldsymbol{x}} - \boldsymbol{x}_{\mathrm{B}}||_2 \leq ||\boldsymbol{x} - \boldsymbol{x}_{\mathrm{B}}||_2 + ||\boldsymbol{x} - \tilde{\boldsymbol{x}}||_2 \leq \left(1 + \frac{1}{\sqrt{3}}\right)\frac{M}{\rho}.$$

Hence, to summarize,

$$f(\boldsymbol{x}_{\mathrm{B}}) - f(\boldsymbol{x}^*) \geq \left(\frac{1}{3} - \frac{1}{6\sqrt{3}}\right)M||\tilde{\boldsymbol{x}} - \boldsymbol{x}_{\mathrm{B}}||_2^2,$$

and the needed result is obtained by applying the above to inequality (31).

Now we prove inequality (8). The proof consists of three steps. For brevity, let $H \triangleq \nabla^2 f(\boldsymbol{x}_{\mathrm{B}})$ and $\boldsymbol{x}_+ = \boldsymbol{x}_{\mathrm{B}} - \hat{H}^{-1}Z^{-1}\hat{\boldsymbol{g}}$. We first prove that

$$\tilde{f}(\boldsymbol{x}) \leq \tilde{f}(\boldsymbol{x}_+) + \mathbb{1}\left(\tilde{f}(\boldsymbol{x}_{\mathrm{B}}) \geq \frac{M^3}{8\rho^2}\right) \cdot \tilde{f}(\boldsymbol{x}_{\mathrm{B}}). \tag{34}$$

We shall repetitively use the fact that $||\boldsymbol{z} - \tilde{\boldsymbol{x}}||_2 \leq \sqrt{2\tilde{f}(\boldsymbol{z})/M}$ for any $\boldsymbol{z} \in \mathbb{R}^d$, which is due to strong convexity. When both $\tilde{f}(\boldsymbol{x}_+)$ and $\tilde{f}(\boldsymbol{x}_{\mathrm{B}})$ are no greater than $\frac{M^3}{8\rho^2}$, both $||\boldsymbol{x}_+ - \tilde{\boldsymbol{x}}||_2$ and $||\boldsymbol{x}_{\mathrm{B}} - \tilde{\boldsymbol{x}}||_2$ are no greater than $\frac{M}{2\rho}$. By triangle inequality, we have $||\boldsymbol{x}_+ - \boldsymbol{x}_{\mathrm{B}}||_2 \leq \frac{M}{\rho}$. Recall the construction of $\boldsymbol{x}$, which is identical to $\boldsymbol{x}_+$ in this case, inequality (34) clearly holds. Otherwise, note that $\boldsymbol{x}$ belongs to the line segment between $\boldsymbol{x}_{\mathrm{B}}$ and $\boldsymbol{x}_+$. By convexity, we always have $\tilde{f}(\boldsymbol{x}) \leq \max\{\tilde{f}(\boldsymbol{x}_+), \tilde{f}(\boldsymbol{x}_{\mathrm{B}})\}$. Recall that in this case, $\tilde{f}(\boldsymbol{x}_{\mathrm{B}}) \geq \tilde{f}(\boldsymbol{x}_+)$ can only hold when $\tilde{f}(\boldsymbol{x}_{\mathrm{B}}) \geq \frac{M^3}{8\rho^2}$, we have $\tilde{f}(\boldsymbol{x}) \leq \mathbb{1}\left(\tilde{f}(\boldsymbol{x}_{\mathrm{B}}) \leq \frac{M^3}{8\rho^2}\right)\tilde{f}(\boldsymbol{x}_+) + \mathbb{1}\left(\tilde{f}(\boldsymbol{x}_{\mathrm{B}}) \geq \frac{M^3}{8\rho^2}\right)\max\{\tilde{f}(\boldsymbol{x}_+), \tilde{f}(\boldsymbol{x}_{\mathrm{B}})\}$, which implies inequality (34).

As the second step, we prove that

$$\mathbb{E}\left[\tilde{f}(\boldsymbol{x}_+) \mid \boldsymbol{x}_{\mathrm{B}}\right] \leq \left(\frac{7d^2\rho^{\frac{4}{3}}}{M^2 n_{\mathrm{H}}^{\frac{1}{3}}} + \frac{26d}{n_{\mathrm{g}}}\right)\tilde{f}(\boldsymbol{x}_{\mathrm{B}}) + \frac{3d\rho^{\frac{2}{3}}}{Mn_{\mathrm{g}}^{\frac{2}{3}}}. \tag{35}$$

Note that the estimation error of $\tilde{\boldsymbol{x}}$ can be decomposed into two terms, i.e.,

$$\boldsymbol{x}_+ - \tilde{\boldsymbol{x}} = H^{-1}\nabla f(\boldsymbol{x}_{\mathrm{B}}) - \hat{H}^{-1}Z^{-1}\hat{\boldsymbol{g}}$$
$$= (H^{-1} - \hat{H}^{-1})\nabla f(\boldsymbol{x}_{\mathrm{B}}) + \hat{H}^{-1}Z^{-1}(Z\nabla f(\boldsymbol{x}_{\mathrm{B}}) - \hat{\boldsymbol{g}}),$$

where the first term is due to the error of the Hessian estimator, and the second is mostly contributed by the GradientEst estimator. We apply the AM-QM inequality to their quadratic forms, i.e.,

$$\tilde{f}(\boldsymbol{x}_+) = \left|\left|H^{\frac{1}{2}}\left((H^{-1} - \hat{H}^{-1})\nabla f(\boldsymbol{x}_{\mathrm{B}}) + \hat{H}^{-1}Z^{-1}(Z\nabla f(\boldsymbol{x}_{\mathrm{B}}) - \hat{\boldsymbol{g}})\right)\right|\right|_2^2$$

$$\leq ||H^{\frac{1}{2}}(H^{-1} - \hat{H}^{-1})\nabla f(\boldsymbol{x}_{\mathrm{B}})||_2^2 + ||H^{\frac{1}{2}}\hat{H}^{-1}Z^{-1}(Z\nabla f(\boldsymbol{x}_{\mathrm{B}}) - \hat{\boldsymbol{g}})||_2^2.$$

$$= ||H^{\frac{1}{2}}(H^{-1} - \hat{H}^{-1})\nabla f(\boldsymbol{x}_{\mathrm{B}})||_2^2 + \left(\frac{\lambda_{Z_H}}{r_{\mathrm{g}}}\right)^2 ||H^{\frac{1}{2}}\hat{H}^{-\frac{1}{2}}||^2 \cdot ||Z\nabla f(\boldsymbol{x}_{\mathrm{B}}) - \hat{\boldsymbol{g}}||_2^2,$$

where $\lambda_{Z_H}$, $r_{\mathrm{g}}$ are defined in Algorithm 4 and $|| \cdot ||$ denotes the spectrum norm. By theorem 4.1, we can first take the expectation of the above bound conditioned on any realization of $\hat{H}$. Specifically,

$$\mathbb{E}\left[||Z\nabla f(\boldsymbol{x}_{\mathrm{B}}) - \hat{\boldsymbol{g}}||_2^2 \mid \hat{H}, \boldsymbol{x}_{\mathrm{B}}\right] \leq \left|\left|Z\nabla f(\boldsymbol{x}_{\mathrm{B}}) - \mathbb{E}\left[\hat{\boldsymbol{g}} \mid \hat{H}, \boldsymbol{x}_{\mathrm{B}}\right]\right|\right|_2^2 + \mathrm{Tr}\left(\mathrm{Cov}\left[\hat{\boldsymbol{g}} \mid \hat{H}, \boldsymbol{x}_{\mathrm{B}}\right]\right)$$

$$\leq \left(\frac{r_{\mathrm{g}}^3\rho\sqrt{d}}{2(d+2)}\right)^2 + \frac{2d}{n_{\mathrm{g}}}||Z\nabla f(\boldsymbol{x}_{\mathrm{B}})||_2^2 + \frac{d^2}{18n_{\mathrm{g}}}\left(\rho r_{\mathrm{g}}^3\right)^2 + \frac{d^2}{2n_{\mathrm{g}}}.$$

Recall the definition of $Z$, we have

$$||Z\nabla f(\boldsymbol{x}_\mathrm{B})||_2^2 = \frac{r_\mathrm{g}^2}{\lambda_{Z_H}^2}||\hat{H}^{-\frac{1}{2}}\nabla f(\boldsymbol{x}_\mathrm{B})||_2^2.$$

Hence, by our choice of $r_\mathrm{g}$ in Algorithm 4 and note that $\lambda_{Z_H} \le M^{-\frac{1}{2}}$ is implied by strong convexity,

$$\mathbb{E}\left[\tilde{f}(\boldsymbol{x}_+) \mid \hat{H}, \boldsymbol{x}_\mathrm{B}\right] \le ||H^{\frac{1}{2}}(H^{-1} - \hat{H}^{-1})\nabla f(\boldsymbol{x}_\mathrm{B})||_2^2$$

$$+ ||H^{\frac{1}{2}}\hat{H}^{-\frac{1}{2}}||^2 \left(\frac{3d\rho^{\frac{2}{3}}}{4Mn_\mathrm{g}^{\frac{2}{3}}}\left(1 + \frac{2d^3}{27n_\mathrm{g}}\right) + \frac{2d}{n_\mathrm{g}}||\hat{H}^{-\frac{1}{2}}\nabla f(\boldsymbol{x}_\mathrm{B})||_2^2\right)$$

$$\le \left(||H^{\frac{1}{2}}(H^{-1} - \hat{H}^{-1})H^{\frac{1}{2}}||^2 + \frac{2d}{n_\mathrm{g}}||H^{\frac{1}{2}}\hat{H}^{-\frac{1}{2}}||^4\right) \cdot ||H^{-\frac{1}{2}}\nabla f(\boldsymbol{x}_\mathrm{B})||_2^2$$

$$+ ||H^{\frac{1}{2}}\hat{H}^{-\frac{1}{2}}||^2 \left(\frac{3d\rho^{\frac{2}{3}}}{4Mn_\mathrm{g}^{\frac{2}{3}}}\left(1 + \frac{2d^3}{27n_\mathrm{g}}\right)\right).$$

To characterize the above bound, we first note that the singular values of $H^{\frac{1}{2}}\hat{H}^{-\frac{1}{2}}$ equals the eigenvalues of $\hat{H}^{-\frac{1}{2}}H\hat{H}^{-\frac{1}{2}} = I_d + \hat{H}^{-\frac{1}{2}}(H - \hat{H})\hat{H}^{-\frac{1}{2}}$. As the eigenvalues of $\hat{H}$ are no less than $M$, by triangle inequality, all eigenvalues of $(I_d + \hat{H}^{-\frac{1}{2}}(H - \hat{H})\hat{H}^{-\frac{1}{2}})$ are bounded within $[1 - \frac{||H-\hat{H}||_\mathrm{F}}{M}, 1 + \frac{||H-\hat{H}||_\mathrm{F}}{M}]$. Hence, we have

$$||H^{\frac{1}{2}}\hat{H}^{-\frac{1}{2}}||^2 \le 1 + \frac{||H - \hat{H}||_\mathrm{F}}{M}.$$

Similarly, bounds on the singular values of $H^{\frac{1}{2}}\hat{H}^{-\frac{1}{2}}$ imply bounds on the eigenvalues of $H^{\frac{1}{2}}\hat{H}^{-1}H^{\frac{1}{2}}$, i.e.,

$$||H^{\frac{1}{2}}(H^{-1} - \hat{H}^{-1})H^{\frac{1}{2}}|| \le \frac{||H - \hat{H}||_\mathrm{F}}{M}.$$

Therefore, we have

$$\mathbb{E}\left[\tilde{f}(\boldsymbol{x}_+) \mid \hat{H}, \boldsymbol{x}_\mathrm{B}\right] \le \left(\left(\frac{||H - \hat{H}||_\mathrm{F}}{M}\right)^2 + \frac{2d}{n_\mathrm{g}}\left(1 + \frac{||H - \hat{H}||_\mathrm{F}}{M}\right)^2\right) \cdot ||H^{-\frac{1}{2}}\nabla f(\boldsymbol{x}_\mathrm{B})||_2^2$$

$$+ \left(1 + \frac{||H - \hat{H}||_\mathrm{F}}{M}\right) \cdot \left(\frac{3d\rho^{\frac{2}{3}}}{4Mn_\mathrm{g}^{\frac{2}{3}}}\left(1 + \frac{2d^3}{27n_\mathrm{g}}\right)\right).$$

Now that the above bound is simply a polynomial of $||H - \hat{H}||_\mathrm{F}$. We can use Theorem 4.3 to obtain $\mathbb{P}\left[\left|\left|\hat{H} - H\right|\right|_\mathrm{F} \ge K\right] \le 2\exp\left(-\frac{K^2}{16d^2\rho^{\frac{4}{3}}/n_\mathrm{H}^{\frac{2}{3}}}\right)$ then apply a direct integration. We utilize the assumptions in the statement of proposition to obtain a simpler estimate, expressed as follows.

$$\mathbb{E}\left[\tilde{f}(\boldsymbol{x}_+) \mid \boldsymbol{x}_\mathrm{B}\right] \le \left(\frac{7d^2\rho^{\frac{4}{3}}}{M^2 n_\mathrm{H}^{\frac{1}{3}}} + \frac{26d}{n_\mathrm{g}}\right) \cdot ||H^{-\frac{1}{2}}\nabla f(\boldsymbol{x}_\mathrm{B})||_2^2 + \frac{3d\rho^{\frac{2}{3}}}{Mn_\mathrm{g}^{\frac{2}{3}}}.$$

Then, inequality (8) is implied by the definition of $\tilde{f}$.

For the third step, we observe that the earlier proof steps imply that

$$\mathbb{E}\left[\tilde{f}(\boldsymbol{x}) \mid \boldsymbol{x}_\mathrm{B}\right] \le \left(\frac{7d^2\rho^{\frac{4}{3}}}{M^2 n_\mathrm{H}^{\frac{1}{3}}} + \frac{26d}{n_\mathrm{g}} + \mathbb{1}\left(\tilde{f}(\boldsymbol{x}_\mathrm{B}) \ge \frac{M^3}{8\rho^2}\right)\right) \cdot \tilde{f}(\boldsymbol{x}_\mathrm{B}) + \frac{3d\rho^{\frac{2}{3}}}{Mn_\mathrm{g}^{\frac{2}{3}}}, \qquad (36)$$

and it remains characterize $\tilde{f}(\boldsymbol{x}_\mathrm{B})$. To that end, we reuse inequality (32) and (33), which implies that $\tilde{f}(\boldsymbol{x}_\mathrm{B}) \le 2(f(\boldsymbol{x}_\mathrm{B}) - f(\boldsymbol{x}^*))$ when $||\tilde{\boldsymbol{x}} - \boldsymbol{x}_\mathrm{B}||_2 \le \frac{3M}{2\rho}$. For the other case, we have $||\tilde{\boldsymbol{x}} - \boldsymbol{x}_\mathrm{B}||_2 \ge \frac{3M}{2\rho}$. We instead let $\boldsymbol{x}_\mathrm{r} \triangleq \boldsymbol{x}_\mathrm{B} + \sqrt{\frac{3M}{2\rho||\tilde{\boldsymbol{x}} - \boldsymbol{x}_\mathrm{B}||_2}}(\tilde{\boldsymbol{x}} - \boldsymbol{x}_\mathrm{B})$ and the Lipschitz Hessian condition implies that

$$f(\boldsymbol{x}^*) \le f(\tilde{\boldsymbol{x}}_\mathrm{r}) \le f(\boldsymbol{x}_\mathrm{B}) - \tilde{f}(\boldsymbol{x}_\mathrm{B}) + \tilde{f}(\boldsymbol{x}_\mathrm{r}) + \frac{1}{6}\rho||\tilde{\boldsymbol{x}} - \boldsymbol{x}_\mathrm{r}||_2^3.$$

By convexity, we have $\tilde{f}(\boldsymbol{x}_{\mathrm{B}}) - \tilde{f}(\boldsymbol{x}_{\mathrm{r}}) \geq \sqrt{\frac{3M}{2\rho||\tilde{\boldsymbol{x}} - \boldsymbol{x}_{\mathrm{B}}||_2}}\tilde{f}(\boldsymbol{x}_{\mathrm{B}})$, which can be applied to the above bound. Then, together with inequality (33) and the condition of $||\tilde{\boldsymbol{x}} - \boldsymbol{x}_{\mathrm{B}}||_2$, we have

$$f(\boldsymbol{x}_{\mathrm{B}}) - f(\boldsymbol{x}^*) \geq \sqrt{\frac{3M}{2\rho||\tilde{\boldsymbol{x}} - \boldsymbol{x}_{\mathrm{B}}||_2}}\left(\tilde{f}(\boldsymbol{x}_{\mathrm{B}}) - \frac{M}{4}||\tilde{\boldsymbol{x}} - \boldsymbol{x}_{\mathrm{B}}||_2^2\right) \tag{37}$$

$$\geq \frac{1}{2}\left(\frac{3M}{2\rho||\tilde{\boldsymbol{x}} - \boldsymbol{x}_{\mathrm{B}}||_2}\right)^{\frac{2}{3}}\tilde{f}(\boldsymbol{x}_{\mathrm{B}})$$

$$\geq \frac{3^{\frac{2}{3}}M}{4\rho^{\frac{2}{3}}}\tilde{f}(\boldsymbol{x}_{\mathrm{B}})^{\frac{2}{3}}.$$

To summarize, the following inequality holds in both cases.

$$\tilde{f}(\boldsymbol{x}_{\mathrm{B}}) \leq \max\left\{2(f(\boldsymbol{x}_{\mathrm{B}}) - f(\boldsymbol{x}^*)), \frac{8\rho}{3M^{\frac{3}{2}}}(f(\boldsymbol{x}_{\mathrm{B}}) - f(\boldsymbol{x}^*))^{\frac{3}{2}}\right\}. \tag{38}$$

To apply inequality (36), we use the following implications.

$$\tilde{f}(\boldsymbol{x}_{\mathrm{B}}) \leq 2(f(\boldsymbol{x}_{\mathrm{B}}) - f(\boldsymbol{x}^*)) + \frac{8\rho}{3M^{\frac{3}{2}}}(f(\boldsymbol{x}_{\mathrm{B}}) - f(\boldsymbol{x}^*))^{\frac{3}{2}},$$

$$\mathbb{1}\left(\tilde{f}(\boldsymbol{x}_{\mathrm{B}}) \geq \frac{M^3}{8\rho^2}\right)\tilde{f}(\boldsymbol{x}_{\mathrm{B}}) \leq \frac{8\rho}{M^{\frac{3}{2}}}(f(\boldsymbol{x}_{\mathrm{B}}) - f(\boldsymbol{x}^*))^{\frac{3}{2}}.$$

Then, the derived inequality can be simplified using our assumptions on $n_{\mathrm{g}}$ and $n_{\mathrm{H}}$.

## E  Remaining details for Theorem 3.1

To complete the proof, we essentially need to prove Theorem 4.4. To illustrate the main ideas, we start with an analysis in a simplified setting where estimation errors for the BootstrapingEst and HessianEst functions are zero. Then, we show how the proof steps can be modified to have the errors and uncertainties incorporated.

### E.1  Analysis for the zero-error case

We prove that when the estimation errors are set to zero, the first stage of Algorithm 4 reduces $\nabla f(\boldsymbol{z}_t)$ to a vector of bounded length in boundedly many iterations. This is summarized in the following proposition.

**Proposition E.1.** *For any fixed parameter values $\rho, M, R$, let $\{\boldsymbol{z}_t\}_{t \in \mathbb{N}_+}$ be sequences defined for any $f \in \mathcal{F}(\rho, M, R)$, such that $\boldsymbol{z}_1 = \boldsymbol{0}$ and $(\boldsymbol{z}_{t+1} - \boldsymbol{z}_t)$ equals $-\tilde{H}_t^{-1}\nabla f(\boldsymbol{z}_t)$, where $\tilde{H}_t$ is a matrix that has the same eigenvectors of $\nabla^2 f(\boldsymbol{z}_t)$, with each eigenvalue $\lambda$ replaced by $\max\{\lambda, m_t\}$, and $m_t$ being the smallest value for $||\tilde{H}_t^{-1}\nabla f(\boldsymbol{z}_t)||_2 \leq \frac{M}{\rho}$. There exists an explicit function $T(\rho, M, R) \leq 5R^2\rho^2/M^2 + 1$ such that $\nabla f(\boldsymbol{z}_t) \leq \frac{M^2}{2\rho}$ holds for any $f$ and any $t \geq T(\rho, M, R)$.*

*Proof.* For convenience, let $\tilde{\boldsymbol{r}}_t \triangleq -(\nabla^2 f(\boldsymbol{z}_t))^{-1}\nabla f(\boldsymbol{z}_t)$ and $\boldsymbol{r}_t \triangleq -\tilde{H}_t^{-1}\nabla f(\boldsymbol{z}_t)$. To investigate the evolution of gradients, we integrate the Lipschitz Hessian condition and obtain that

$$||\nabla f(\boldsymbol{z}_{t+1}) - \nabla f(\boldsymbol{z}_t) - \nabla^2 f(\boldsymbol{z}_t)\,\boldsymbol{r}_t||_2 \leq \frac{1}{2}\rho||\boldsymbol{r}_t||_2^2. \tag{39}$$

From the definition of $\boldsymbol{r}_t$, if $||\tilde{\boldsymbol{r}}_t||_2 \leq M/\rho$, we have

$$||\nabla f(\boldsymbol{z}_{t+1})||_2 \leq \frac{1}{2}\rho||\tilde{\boldsymbol{r}}_t||_2^2 \leq \frac{M^2}{2\rho}. \tag{40}$$

Note that by strong convexity, when the above bound holds,

$$||\tilde{\boldsymbol{r}}_{t+1}||_2 \leq \frac{||\nabla f(\boldsymbol{z}_{t+1})||_2}{M} \leq \frac{M}{2\rho} \leq \frac{M}{\rho}. \tag{41}$$

Hence, once $||\tilde{r}_t||_2$ reaches below $M/\rho$ for some $t_0$, our desired bound on $||\nabla f(z_{t+1})||_2$ remain hold for any $t > t_0$. Therefore, for the purpose of our proof, we can focus on the case where $||\tilde{r}_t||_2 > M/\rho$ and show that this condition can only hold for boundedly many iterations.

Consider any fixed function $f \in \mathcal{F}(\rho, M, R)$, let $x^*$ denote its global minimum point. A crucial step in our proof is to show that

$$(x^* - z_t) \cdot r_t \geq 0.6||r_t||_2^2. \tag{42}$$

For brevity, let $\beta$ denote the minimum eigenvalue of $\tilde{H}_t$, and $\tilde{f}$ denote the following quadratic approximation.

$$\tilde{f}(x) \triangleq f(z_t) + (x - z_t)\nabla f(z_t) + \frac{1}{2}(x - z_t)(\nabla^2 f(z_t))(x - z_t).$$

We apply the strong convexity condition at point $y \triangleq z_{t+1} - 0.4\beta\tilde{H}_t^{-1}r_t$. Recall that $f$ is minimized at $x^*$, we have

$$0 \geq f(x^*) - f(y) \geq \nabla f(y) \cdot (x^* - y) + \frac{M}{2}||x^* - y||_2^2. \tag{43}$$

By integrating the Lipschitz Hessian, similar to inequality (39), $\nabla f(y)$ can be approximated with $\nabla f(z_t) + \nabla^2 f(z_t)(y - z_t) = \nabla \tilde{f}(y)$. Formally,

$$||\nabla f(y) - \nabla \tilde{f}(y)||_2 \leq \frac{1}{2}\rho||y - z_t||_2^2. \tag{44}$$

Hence, inequality (43) implies that

$$0 \geq \nabla \tilde{f}(y) \cdot (x^* - y) + \frac{M}{2}||x^* - y||_2^2 - \frac{1}{2}\rho||y - z_t||_2^2||x^* - y||_2. \tag{45}$$

To characterize the terms in the above inequality, we first note that

$$||y - z_t||_2^2 = ||r_t||_2^2 - 0.8r_t \cdot \tilde{H}_t^{-1}\beta r_t + 0.16||\tilde{H}_t^{-1}\beta r_t||_2^2$$
$$\leq ||r_t||_2^2 - 0.64||\tilde{H}_t^{-1}\beta r_t||_2^2.$$

For convenience, we denote $c \triangleq ||\tilde{H}_t^{-1}\beta r_t||_2/||r_t||_2$. We have that $c \in (0, 1]$ and

$$||y - z_t||_2^2 \leq \left(1 - 0.64c^2\right)||r_t||_2^2. \tag{46}$$

We also consider the following vector,

$$q \triangleq (\nabla^2 f(z_t))^{-1}\left(\nabla \tilde{f}(y) + (\beta + 0.36cM)r_t - 0.6cM(y - z_t)\right)$$
$$= 0.6\tilde{H}_t^{-1}\left(\beta r_t + 0.4cM\left(\tilde{H}_t^{-1}\beta r_t - r_t\right)\right),$$

of which the L2 norm is no greater than $0.6c||r_t||_2 \leq 0.6cM/\rho$, which can be proved in the eigenbasis of $\nabla^2 f(z_t)$. By Cauchy's inequality, we have that

$$q \cdot \nabla \tilde{f}(x^*) \geq -||q||_2||\nabla \tilde{f}(x^*)||_2$$
$$\geq -\frac{0.6cM}{\rho}||\nabla \tilde{f}(x^*)||_2. \tag{47}$$

Note that $x^* - y = (\nabla^2 f(z_t))^{-1}\left(\nabla \tilde{f}(x^*) - \nabla \tilde{f}(y)\right)$. The LHS of the above inequality can be written as $(x^* - y) \cdot (\nabla^2 f(z_t)) \cdot q + q \cdot \nabla \tilde{f}(y)$, where the first term contains $\nabla \tilde{f}(y) \cdot (x^* - y)$, and the second term is bounded as follows.

$$q \cdot \nabla \tilde{f}(y) = -0.6\tilde{H}_t^{-1}\left(\tilde{H}_t^{-1}\beta^2 r_t + (\beta - 0.4cM)(r_t - \tilde{H}_t^{-1}\beta r_t)\right)$$
$$\cdot \left(0.4\beta r_t + 0.6\left(\tilde{H}_t - \nabla^2 f(z_t)\right)r_t\right)$$
$$\leq -0.6\tilde{H}_t^{-1} \cdot \tilde{H}_t^{-1}\beta^2 r_t \cdot 0.4\beta r_t$$
$$= -0.24c^2\beta||r_t||_2^2. \tag{48}$$

On the other hand, observe that inequality (44) holds for any generic $\boldsymbol{y} \in \mathbb{R}^d$. The RHS of inequality (47) can be characterized as follows.

$$
\begin{aligned}
||\nabla \tilde{f}(\boldsymbol{x}^*)||_2 &= ||\nabla f(\boldsymbol{x}^*) - \nabla \tilde{f}(\boldsymbol{x}^*)||_2 \\
&\leq \frac{1}{2}\rho ||\boldsymbol{x}^* - \boldsymbol{z}_t||_2^2 \\
&= \frac{1}{2}\rho ||\boldsymbol{x}^* - \boldsymbol{y}||_2^2 + \rho(\boldsymbol{x}^* - \boldsymbol{y}) \cdot (\boldsymbol{y} - \boldsymbol{z}_t) + \frac{1}{2}\rho ||\boldsymbol{y} - \boldsymbol{z}_t||_2^2. \quad (49)
\end{aligned}
$$

Therefore, by combining inequalities (45), (47), and (49), we have that

$$
\begin{aligned}
(\boldsymbol{x}^* - \boldsymbol{y}) \cdot (\beta + 0.36cM)\, \boldsymbol{r}_t \geq\ & -\boldsymbol{q} \cdot \nabla \tilde{f}(\boldsymbol{y}) + (0.5 - 0.3c)M ||\boldsymbol{x}^* - \boldsymbol{y}||_2^2 \\
& - 0.5\rho ||\boldsymbol{y} - \boldsymbol{z}_t||_2^2 \cdot ||\boldsymbol{x}^* - \boldsymbol{y}||_2 - 0.3cM ||\boldsymbol{y} - \boldsymbol{z}_t||_2^2 \\
\geq\ & -\boldsymbol{q} \cdot \nabla \tilde{f}(\boldsymbol{y}) - \frac{\rho^2 ||\boldsymbol{y} - \boldsymbol{z}_t||_2^4}{(8 - 2.4c)M} - 0.3cM ||\boldsymbol{y} - \boldsymbol{z}_t||_2^2,
\end{aligned}
$$

where the second line is obtained by taking the infimum w.r.t. $||\boldsymbol{x}^* - \boldsymbol{y}||_2$. Then, we apply inequalities (46), (48), and $||\boldsymbol{r}_t||_2 \leq M/\rho$ to obtain the following bound.

$$
(\boldsymbol{x}^* - \boldsymbol{y}) \cdot \boldsymbol{r}_t \geq \frac{0.24c^2\beta - \frac{M(1 - 0.64c^2)^2}{(8 - 2.4c)} - 0.3cM(1 - 0.64c^2)}{\beta + 0.36cM} \cdot ||\boldsymbol{r}_t||_2^2.
$$

Note that the above bound is non-decreasing w.r.t. $\beta$, and our construction implies $\beta \geq M$. We can substitute $\beta$ in the above inequality with $M$. Further, note that

$$
\begin{aligned}
(\boldsymbol{y} - \boldsymbol{z}_t) \cdot \boldsymbol{r}_t &= ||\boldsymbol{r}_t||_2^2 - \boldsymbol{r}_t \cdot 0.4\beta \tilde{H}_t^{-1} \boldsymbol{r}_t \\
&\geq (1 - 0.4c) ||\boldsymbol{r}_t||_2.
\end{aligned}
$$

We have obtained a lower bound of $(\boldsymbol{x}^* - \boldsymbol{z}_t) \cdot \boldsymbol{r}_t$ as a function of $c$. This dependency is removed by taking the infimum, i.e.,

$$
(\boldsymbol{x}^* - \boldsymbol{z}_t) \cdot \boldsymbol{r}_t \geq \inf_{c \in (0,1]} \left( \frac{0.24c^2 - \frac{(1 - 0.64c^2)^2}{(8 - 2.4c)} - 0.3c(1 - 0.64c^2)}{1 + 0.36c} + 1 - 0.4c \right) \cdot ||\boldsymbol{r}_t||_2,
$$

then inequality (42) is obtained.

We use this key inequality to obtain the following recursion rule.

$$
||\boldsymbol{x}^* - \boldsymbol{z}_t||_2^2 - ||\boldsymbol{x}^* - \boldsymbol{z}_{t+1}||_2^2 = 2\boldsymbol{r}_t \cdot (\boldsymbol{x}^* - \boldsymbol{z}_t) - ||\boldsymbol{r}_t||_2^2 \geq 0.2 ||\boldsymbol{r}_t||_2^2.
$$

Recall that for $f \in \mathcal{F}(\rho, M, R)$, we assumed that $||\boldsymbol{x}^*||_2 \leq R$. Therefore, for $\boldsymbol{z}_1 = \boldsymbol{0}$, the above recursion implies that the inequality $||\tilde{\boldsymbol{r}}_t||_2 > M/\rho$ can hold for no greater than $5R^2\rho^2/M^2$ iterations as $||\boldsymbol{x}^* - \boldsymbol{z}_t||_2^2$ has to be non-negative for any $t$. Hence, based on the earlier discussion, we have proved that either $||\tilde{\boldsymbol{r}}_t||_2 \leq M/\rho$ or $\nabla f(\boldsymbol{z}_{t+1}) = \boldsymbol{0}$ for all $t \geq 5R^2\rho^2/M^2$ and all $f \in \mathcal{F}(\rho, M, R)$. Recall inequality (40), this implies that $||\nabla f(\boldsymbol{z}_t)||_2 \leq \frac{M^2}{2\rho}$ for all $t \geq 5R^2\rho^2/M^2 + 1$. $\quad\square$

**Remark E.2.** *Recall the recursion provided by inequality (40) and (41). The gradients for the sequence $\boldsymbol{z}_t$ decay double-exponentially once they are sufficiently close to zero. Hence, Proposition E.1 proves that it takes finitely many iterations for the bootstrapping stage of Algorithm 4 to get arbitrarily close to $\boldsymbol{x}^*$ in the zero-error case.*

### E.2 Generalization to the noisy case

Now we prove that, given a bounded number of iterations, the bootstrapping stage in Algorithm 4 provides an $\boldsymbol{x}_{\mathrm{B}}$ that is sufficiently close to $\boldsymbol{x}^*$ with high probability even in the presence of noise. Similar to the zero-error case, we provide the following guarantee.

**Theorem E.3.** *For any fixed $\rho$, $M$ and $R$, the result returned by the first stage of Algorithm 4 satisfies*

$$
\lim_{T \to \infty} \sup_{f \in \mathcal{F}(\rho, M, R)} \mathbb{E}\left[ \left|\left|\nabla f(\boldsymbol{x}_N^{(\mathrm{B})})\right|\right|_2^3 \cdot T^{\frac{2}{3}} \right] = 0. \quad (50)
$$

For convenience, let $x_k^{(B)}$ denote the realization of vector $x$ at the end of the $k$th iteration in the bootstrapping stage and $N \triangleq \lfloor T^{0.1} \rfloor$ denote the number of iterations. Therefore, we have $x_B = x_N^{(B)}$. Further, we define $x_0^{(B)} \triangleq \mathbf{0}$. We let $m_k, H_k$ denote the realization of $\hat{m}, H_{m^*}$ in the $(k+1)$th iteration of the bootstrapping stage. Therefore, we have $x_{k+1}^{(B)} = x_k^{(B)} - H_k^{-1} m_k$. As a Benchmark for our analysis, we use $\tilde{H}_k$ to denote the value of $H_{m^*}$ in the zero-error case, i.e., they denotes the value of $H_{m^*}$ under the special case of $\hat{m} = \nabla f\left(x_k^{(B)}\right)$ and $\hat{H} = \nabla^2 f\left(x_k^{(B)}\right)$. Hence, the update in the zero-error case can be denoted as $r_k \triangleq -\tilde{H}_k^{-1} \nabla f\left(x_k^{(B)}\right)$.

We let $E_k$ be the indicator function of the event where there exists an $j < k$ such that $\left\|x_{j+1}^{(B)} - x_j^{(B)} - r_j\right\|_2 \geq MT^{-0.2}/\rho$. Intuitively, $E_k = 0$ describes the event that the optimization steps can be characterized similar to the zero-error case. Notice that $E_k$ is non-decreasing. We have either $E_N = 0$, or $E_{k_0} = 1$ for some $k_0 \in \{1, 2, ..., N\}$. We provide the analysis of Theorem E.3 separately for each of these two cases.

For the first case, i.e., when $E_N = 0$, we can follow the earlier arguments and prove the following proposition (see Appendix F.3 for details).

**Proposition E.4.** *For any function $f \in \mathcal{F}(\rho, M, R)$ and any sequence $x_0^{(B)}, x_1^{(B)}, ..., x_{N-1}^{(B)} \in \mathbb{R}^d$ that satisfies $x_0^{(B)} = \mathbf{0}$ and $E_{N-1} = 0$, we have $\|r_{N-1}\|_2 \leq 2MT^{-0.2}/\rho$ when $T$ is sufficiently large.*

Recall the definition of $E_N = 0$. The above proposition immediately implies that

$$\left\|x_N^{(B)} - x_{N-1}^{(B)}\right\|_2 \leq 3MT^{-0.2}/\rho < M/\rho$$

when $T$ is large. Hence, in such cases, $x_N^{(B)}$ is obtained by the Newton update. Formally, if $\hat{H}$ denotes the estimator returned by the HessianEst function in the $N$th iteration, we have

$$(x_N^{(B)} - x_{N-1}^{(B)}) \cdot \hat{H} = -m_{N-1}, \tag{51}$$

Therefore, by applying the above results to the Lipschitz Hessian condition, we have

$$\left\|\nabla f(x_N^{(B)})\right\|_2 \leq \left\|\nabla f(x_{N-1}^{(B)}) + (x_N^{(B)} - x_{N-1}^{(B)}) \cdot \nabla^2 f(x_{N-1}^{(B)})\right\|_2 + \frac{\rho}{2}\|x_N^{(B)} - x_{N-1}^{(B)}\|_2^2$$

$$\leq \left\|\nabla f(x_{N-1}^{(B)}) - m_{N-1} + (x_N^{(B)} - x_{N-1}^{(B)}) \cdot \left(\nabla^2 f(x_{N-1}^{(B)}) - \hat{H}\right)\right\|_F + \frac{9M^2}{2\rho T^{0.4}}$$

$$\leq \left\|\nabla f(x_{N-1}^{(B)}) - m_{N-1}\right\|_2 + \frac{3M}{\rho T^{0.2}} \cdot \left\|\nabla^2 f(x_{N-1}^{(B)}) - \hat{H}\right\|_F + \frac{9M^2}{2\rho T^{0.4}}. \tag{52}$$

Hence, by direct integration of the tail bounds in Theorem 4.3, we can conclude that

$$\limsup_{T \to \infty} \sup_{f \in \mathcal{F}(\rho, M, R)} \mathbb{E}\left[\left\|\nabla f(x_N^{(B)})\right\|_2^3 \cdot \mathbb{1}(E_N = 0) \cdot T^{\frac{2}{3}}\right] = 0. \tag{53}$$

Now we consider the second case, i.e., when $E_N = 1$. By its definition, we must have the event of $E_k = 0$ to $E_{k+1} = 1$ for a unique $k \in \{0, 1, ..., N-1\}$, which implies that $\left\|x_{k+1}^{(B)} - x_k^{(B)} - r_k\right\|_2 \geq MT^{-0.2}/\rho$. We prove that conditioned on any of these events, the random variable $\|\nabla f(x_{k+1}^{(B)})\|_2$ has a super-polynomial tail, which contributes vanishingly to their moments in the asymptotic sense. Formally, let

$$M_k \triangleq \mathbb{E}\left[\|\nabla f(x_{k+1}^{(B)})\|_2^3 \cdot \mathbb{1}(E_{k+1} = 1, E_k = 0)\right],$$

We aim to prove that

$$\limsup_{T \to \infty} \max_{k \in \{0, 1, ..., N-1\}} \sup_{f \in \mathcal{F}(\rho, M, R)} M_k \cdot NT^{\frac{2}{3}} = 0. \tag{54}$$

Consider any fixed $k \in \{0, 1, ..., N-1\}$ and conditioned on any realization of $x_k^{(B)}$, we characterize the distribution of $x_{k+1}^{(B)}$ by providing the following proposition, which is proved in Appendix F.4.

**Proposition E.5.** *Consider any vectors $\boldsymbol{m}, \boldsymbol{m}' \in \mathbb{R}^n$, any positive definite matrices $H, H' \in \mathbb{R}^n$ with all eigenvalues lower bounded by $M$, and any fixed parameter $R_0 \in \mathbb{N}_+$. Let $H_{m^*}$ be the symmetric matrix sharing the same eigenbasis of $H$ but with each eigenvalue $\lambda$ replaced with $\max\{\lambda, m^*\}$, where $m^*$ is chosen to be the smallest value such that $||H_{m^*}^{-1}\boldsymbol{m}||_2 \leq R_0$. Let $H'_{m'^*}$ be defined correspondingly for $\boldsymbol{m}'$ and $H'$. We have that*

$$\left|\left|H_{m^*}^{-1}\boldsymbol{m} - H_{m'^*}'^{-1}\boldsymbol{m}'\right|\right|_2^2 \leq \frac{2R_0}{M} \cdot \left(||\boldsymbol{m} - \boldsymbol{m}'||_2 + R_0 \cdot ||H - H'||_{\mathrm{F}}\right). \tag{55}$$

*Furthermore, when $\left|\left|H_{m^*}^{-1}\boldsymbol{m} - H_{m'^*}'^{-1}\boldsymbol{m}'\right|\right|_2 > 0$, we have*

$$\left|\left|H'_{m'^*}\left(H_{m^*}^{-1}\boldsymbol{m} - H_{m'^*}'^{-1}\boldsymbol{m}'\right)\right|\right|_2$$
$$\leq \left(3 + \frac{2R_0}{||H_{m^*}^{-1}\boldsymbol{m} - H_{m'^*}'^{-1}\boldsymbol{m}'||_2}\right)\left(||\boldsymbol{m} - \boldsymbol{m}'||_2 + R_0 ||H - H'||_{\mathrm{F}}\right). \tag{56}$$

By choosing $R_0 = M/\rho$, $H_{m^*} = H_{N-1}$, $H'_{m'^*} = \nabla^2 f(\boldsymbol{x}_{N-1}^{(\mathrm{B})})$, $\boldsymbol{m} = \boldsymbol{m}_{N-1}$, and $\boldsymbol{m}' = \nabla f(\boldsymbol{x}_{N-1}^{(\mathrm{B})})$ for Proposition E.5, the condition of $E_{k+1}$ can be characterized by the estimation errors of the gradient and Hessian. For brevity, we define

$$\Psi \triangleq ||\boldsymbol{m} - \boldsymbol{m}'||_2 + R_0 ||H - H'||_{\mathrm{F}}.$$

We also let $\beta$ denote the minimum eigenvalue of $H_k$. The condition of $E_{k+1} = 1$ and $E_k = 0$ implies that $||H_{m^*}^{-1}\boldsymbol{m} - H_{m'^*}'^{-1}\boldsymbol{m}'||_2 \geq R_0 T^{-0.2}$, which implies that $\Psi \geq \frac{\beta R_0 T^{-0.2}}{3 + 2T^{0.2}}$ according to inequality (56). Hence, $M_k$ can be bounded as follows.

$$M_k \leq \mathbb{E}\left[||\nabla f(\boldsymbol{x}_{k+1}^{(\mathrm{B})})||_2^3 \cdot \mathbb{1}\left(\Psi \geq \frac{\beta R_0 T^{-0.2}}{3 + 2T^{0.2}}\right)\right],$$

On the other hand, by generalizing inequality (52), we have

$$\left|\left|\nabla f(\boldsymbol{x}_{k+1}^{(\mathrm{B})})\right|\right|_2 \leq \left|\left|\nabla f(\boldsymbol{x}_k^{(\mathrm{B})}) + (\boldsymbol{x}_{k+1}^{(\mathrm{B})} - \boldsymbol{x}_k^{(\mathrm{B})}) \cdot \nabla^2 f(\boldsymbol{x}_k^{(\mathrm{B})})\right|\right|_2 + \frac{\rho}{2}||\boldsymbol{x}_{k+1}^{(\mathrm{B})} - \boldsymbol{x}_k^{(\mathrm{B})}||_2^2$$
$$\leq \left|\left|\nabla f(\boldsymbol{x}_k^{(\mathrm{B})}) - \boldsymbol{m}_k\right|\right|_2 + \left|\left|\boldsymbol{x}_{k+1}^{(\mathrm{B})} - \boldsymbol{x}_k^{(\mathrm{B})}\right|\right|_2 \cdot \left|\left|\nabla^2 f(\boldsymbol{x}_k^{(\mathrm{B})}) - \hat{H}\right|\right|_{\mathrm{F}}$$
$$+ \left|\left|\left(\boldsymbol{x}_{k+1}^{(\mathrm{B})} - \boldsymbol{x}_k^{(\mathrm{B})}\right)\left(\hat{H} - H_k\right)\right|\right|_2 + \frac{\rho}{2}||\boldsymbol{x}_{k+1}^{(\mathrm{B})} - \boldsymbol{x}_k^{(\mathrm{B})}||_2^2$$
$$\leq \Psi + R_0\beta + \frac{R_0 M}{2}. \tag{57}$$

Therefore,

$$\limsup_{T \to \infty} \sup_{f \in \mathcal{F}(\rho, M, R)} M_k \cdot NT^{\frac{2}{3}}$$
$$\leq \limsup_{T \to \infty} \mathbb{E}\left[\left(\Psi + R_0\beta + \frac{R_0 M}{2}\right)^3 \cdot \mathbb{1}\left(\Psi \geq \frac{\beta R_0 T^{-0.2}}{3 + 2T^{0.2}}\right) \cdot NT^{\frac{2}{3}}\right]$$
$$= 0. \tag{58}$$

Since the above bounds are uniform over the index $k$, equation (54) is implied. The above arguments also show that

$$\limsup_{T \to \infty} \sup_{f \in \mathcal{F}(\rho, M, R)} \mathbb{P}[E_N = 1] \cdot N^3 T^{\frac{2}{3}}$$
$$\leq \limsup_{T \to \infty} \sup_{f \in \mathcal{F}(\rho, M, R)} N \cdot \max_k \mathbb{P}[E_{k+1} = 1, E_k = 0] \cdot N^3 T^{\frac{2}{3}} = 0. \tag{59}$$

So far, we have proved that the moments of the gradient norm $\left|\left|\nabla f(\boldsymbol{x}_k^{(\mathrm{B})})\right|\right|_2$ is bounded after entering the $E_k = 1$ phase. We proceed to bound their contribution to the $N$th iteration. To that end, we denote

$$G_k \triangleq \mathbb{E}\left[||\nabla f(\boldsymbol{x}_k^{(\mathrm{B})})||_2^3 \cdot \mathbb{1}(E_k = 1)\right].$$

This sequence is initialized with $G_0 = 0$ by definition. We establish the following recursion for sufficiently large $T$.

$$G_{k+1} \le G_k \left(1 + \frac{1}{N}\right) + 6N^2(\rho R_0^2)^3 \cdot \mathbb{P}[E_N = 1] + M_k.$$

We note that conditioned on any fixed $\boldsymbol{x}_k^{(\mathrm{B})}$ the gradient norm function $||\nabla f(\boldsymbol{x}_{k+1}^{(\mathrm{B})})||_2$ can be approximated with its linear expansion. Formally, let $\tilde{g}(\boldsymbol{x}) \triangleq \nabla f(\boldsymbol{x}_k^{(\mathrm{B})}) + (\boldsymbol{x} - \boldsymbol{x}_k^{(\mathrm{B})}) \cdot \nabla^2 f(\boldsymbol{x}_k^{(\mathrm{B})})$, we have

$$\left\lVert \nabla f(\boldsymbol{x}_{k+1}^{(\mathrm{B})}) \right\rVert_2 \le \left\lVert \tilde{g}(\boldsymbol{x}_{k+1}^{(\mathrm{B})}) \right\rVert_2 + \frac{1}{2}\rho \left\lVert \boldsymbol{x}_{k+1}^{(\mathrm{B})} - \boldsymbol{x}_k^{(\mathrm{B})} \right\rVert_2^2$$
$$\le \left\lVert \tilde{g}(\boldsymbol{x}_{k+1}^{(\mathrm{B})}) \right\rVert_2 + \frac{1}{2}\rho R_0^2.$$

Then, in the eigenbasis of $\hat{H}$, it is clear that

$$\left\lVert \tilde{g}(\boldsymbol{x}_{k+1}^{(\mathrm{B})}) \right\rVert_2 \le \left\lVert \nabla f(\boldsymbol{x}_k^{(\mathrm{B})}) + (\boldsymbol{x}_{k+1}^{(\mathrm{B})} - \boldsymbol{x}_k^{(\mathrm{B})}) \cdot \hat{H} \right\rVert_2$$
$$+ \left\lVert (\boldsymbol{x}_{k+1}^{(\mathrm{B})} - \boldsymbol{x}_k^{(\mathrm{B})}) \cdot \left(\hat{H} - \nabla^2 f(\boldsymbol{x}_k^{(\mathrm{B})})\right) \right\rVert_2$$
$$\le \left\lVert \nabla f(\boldsymbol{x}_k^{(\mathrm{B})}) \right\rVert_2 + \left\lVert \boldsymbol{m}_k - \nabla f(\boldsymbol{x}_k^{(\mathrm{B})}) \right\rVert_2 + R_0 \left\lVert \hat{H} - \nabla^2 f(\boldsymbol{x}_k^{(\mathrm{B})}) \right\rVert_{\mathrm{F}}.$$

Recall that by Theorem 4.3, when $T$ is sufficiently large, the moments of $\left\lVert \boldsymbol{m}_k - \nabla f(\boldsymbol{x}_k^{(\mathrm{B})}) \right\rVert_2 + R_0 \left\lVert \hat{H} - \nabla^2 f(\boldsymbol{x}_k^{(\mathrm{B})}) \right\rVert_{\mathrm{F}}$ is upper bounded by any fixed quantity. Therefore, as a rough estimate, we have

$$\mathbb{E}\left[ ||\nabla f(\boldsymbol{x}_{k+1}^{(\mathrm{B})})||_2^3 \cdot \mathbb{1}(E_k = 1) \right] \le \mathbb{E}\left[ ||\nabla f(\boldsymbol{x}_k^{(\mathrm{B})}) + \rho R_0^2||_2^3 \cdot \mathbb{1}(E_k = 1) \right]$$
$$\le G_k \left(1 + \frac{1}{N}\right) + 6N^2(\rho R_0^2)^3 \cdot \mathbb{P}[E_k = 1]$$

when $T$ is sufficiently large. Consequently, our needed recursion is implied by the monotonicity of $E_k$, and we have

$$\limsup_{T \to \infty} \sup_{f \in \mathcal{F}(\rho, M, R)} G_N \cdot T^{\frac{2}{3}}$$
$$\le \limsup_{T \to \infty} \sup_{f \in \mathcal{F}(\rho, M, R)} \left( \max_k M_k + 6N^2(\rho R_0^2)^3 \cdot \mathbb{P}[E_k = 1] \right) \cdot N \left(1 + \frac{1}{N}\right)^N T^{\frac{2}{3}}$$
$$= 0. \tag{60}$$

Finally, Theorem E.3 is proved by noting that

$$\mathbb{E}\left[ \left\lVert \nabla f(\boldsymbol{x}_N^{(\mathrm{B})}) \right\rVert_2^3 \right] = \mathbb{E}\left[ \left\lVert \nabla f(\boldsymbol{x}_N^{(\mathrm{B})}) \right\rVert_2^3 \cdot \mathbb{1}(E_N = 0) \cdot T^{\frac{2}{3}} \right] + G_N. \tag{61}$$

Hence, equation (50) is implied by equation (53) and inequality (60).

### E.3  Proof of Theorem 4.4

*Proof.* Given Theorem E.3, our needed inequality (6) is implied by the strong convexity assumption. Particularly, the implication is due to the fact that $\frac{||\nabla f(x)||_2^2}{2M} \ge f(x) - f^*$ for any $x \in \mathbb{R}^d$.  $\square$

**Remark E.6.** *Note that compared to the simple regret guarantee stated in inequality (6), we have essentially proved a stronger statement that the moments of the gradient at the outcome of the bootstrapping stage follow similar power decay laws. Therefore, while we presented a final stage algorithm that uses non-isotropic sampling to be compatible with general bootstrapping stages, our specific bootstrapping stage actually allows for the use of isotropic (hyperspherical) sampling for gradient estimation in the final stage.*

# F  Proofs of some useful propositions

## F.1  Proof of Proposition C.1

*Proof.* Recall that all $z_j$'s have zero expectations. By subgaussianity, we have that all even moments of $z_j$ are bounded as follows.

$$
\begin{aligned}
\mathbb{E}\left[z_j^{2\ell}\right] &= \int_{K=0}^{+\infty} 2\ell K^{2\ell-1}\mathbb{P}\left[|z_j| \geq K\right] \mathrm{d}K \\
&\leq \int_{K=0}^{+\infty} 2\ell K^{2\ell-1} \min\left\{2\exp\left(-\frac{K^2}{\sigma_j^2}\right), 1\right\} \mathrm{d}K \\
&\leq \begin{cases} (1+\ln 2)\,\sigma_j^2 & \text{if } \ell = 1, \\ (2+2\ln 2 + \ln^2 2)\,\sigma_j^4 & \text{if } \ell = 2, \\ 2\cdot\ell!\,\sigma_j^{2\ell} & \text{if } \ell > 2. \end{cases}
\end{aligned}
\tag{62}
$$

Using AM-GM inequality, the odd moments of $z_j$ can then be bounded using the even moments. Specifically,

$$
\mathbb{E}\left[z_j^{2\ell+1}\right] \leq \frac{1}{2s}\mathbb{E}\left[z_j^{2\ell}\right] + \frac{s}{2}\mathbb{E}\left[z_j^{2\ell+2}\right].
$$

Therefore, we have obtained the following upper bounds for the moment-generating function.

$$
\begin{aligned}
\mathbb{E}\left[\exp(sz_j)\right] &= 1 + \sum_{m=2}^{\infty} \frac{s^m}{m!}\mathbb{E}\left[z_j^m\right] \\
&\leq 1 + \frac{7s^2}{12}\mathbb{E}\left[z_j^2\right] + \sum_{\ell=2}^{\infty}\left(2\ell+2+\frac{1}{2\ell+1}\right)\frac{s^{2\ell}}{(2\ell)!\cdot 2}\mathbb{E}\left[z_j^{2\ell}\right].
\end{aligned}
$$

Applying inequality (62), the expression above can be bounded with a series of $(s\sigma_j)^2$. The coefficient of each $(s\sigma_j)^{2\ell}$ is no greater than $\frac{1}{\ell!}$, which can be verified numerically for $\ell \leq 2$ and inductively for $\ell \geq 3$. Hence, we have

$$
\mathbb{E}\left[\exp(sz_j)\right] \leq \sum_{\ell=0}^{\infty} \frac{(s\sigma_j)^{2\ell}}{\ell!} = e^{(s\sigma_j)^2}.
\tag{63}
$$

Because $z_j$'s are independent,

$$
\mathbb{E}\left[\exp\left(s\sum_j z_j\right)\right] = \prod_j \mathbb{E}\left[\exp(sz_j)\right] \leq \exp\left(s^2\sum_j \sigma_j^2\right).
$$

Inequality (16) is implied by Markov's bound. Specifically, for any $K \geq 0$,

$$
\begin{aligned}
\mathbb{P}\left[\sum_{j=1}^{k} z_j \geq K\right] &\leq \inf_{s\geq 0}\mathbb{E}\left[\exp\left(s\sum_j z_j\right)\right]\cdot\exp\left(-sK\right) \\
&\leq \inf_{s\geq 0}\exp\left(s^2\sum_j \sigma_j^2 - sK\right) \\
&= \exp\left(-\frac{K^2}{4\sum_j \sigma_j^2}\right).
\end{aligned}
$$

For the same reason, we also have

$$
\mathbb{P}\left[\sum_{j=1}^{k} z_j \leq -K\right] \leq \exp\left(-\frac{K^2}{4\sum_j \sigma_j^2}\right).
$$

Hence, by union bound,

$$\mathbb{P}\left[\left|\sum_{j=1}^{k} z_j\right| \geq K\right] \leq \mathbb{P}\left[\sum_{j=1}^{k} z_j \geq K\right] + \mathbb{P}\left[\sum_{j=1}^{k} z_j \leq -K\right] \leq 2\exp\left(-\frac{K^2}{4\sum_j \sigma_j^2}\right). \tag{64}$$

$\square$

## F.2 Proof of Proposition C.2

*Proof.* By subexponentiality, the moment-generating function of each $|z_j|$ is bounded as follows for any $s < \frac{1}{\sigma_j}$.

$$\begin{aligned}
\mathbb{E}[\exp(s|z_j|)] &= 1 + \int_{K=0}^{+\infty} s\exp(sK) \cdot \mathbb{P}[|z_j| \geq K]\, dK \\
&\leq 1 + \int_{K=0}^{+\infty} s\exp(sK) \cdot \min\left\{2\exp\left(-\frac{K}{\sigma_j}\right), 1\right\} dK \\
&= \frac{2^{s\sigma_j}}{1 - s\sigma_j}.
\end{aligned} \tag{65}$$

Because $z_j$'s are independent,

$$\begin{aligned}
\mathbb{E}\left[\exp\left(s\left|\sum_j z_j\right|\right)\right] &\leq \mathbb{E}\left[\exp\left(s\sum_j |z_j|\right)\right] = \prod_j \mathbb{E}[\exp(s|z_j|)] \\
&\leq \frac{2^{s\sum_j \sigma_j}}{\prod_j (1 - s\sigma_j)}.
\end{aligned} \tag{66}$$

We choose $s = 1/(3\sum_j \sigma_j)$, note that $s\sigma_j \leq 1/3$, we have $(1 - s\sigma_j) \geq \left(\frac{2}{3}\right)^{3s\sigma_j}$. Hence,

$$\mathbb{E}\left[\exp\left(s\left|\sum_j z_j\right|\right)\right] \leq e^{(\ln 2 - 3\ln\frac{2}{3})(s\sum_j \sigma_j)} = 3/2^{\frac{2}{3}} < 2.$$

Then, inequality (18) is implied by Markov's bound, i.e.,

$$\begin{aligned}
\mathbb{P}\left[\left|\sum_{j=1}^{k} z_j\right| \geq K\right] &\leq \mathbb{E}\left[\exp\left(s\left|\sum_j z_j\right|\right)\right] \cdot \exp(-sK) \\
&\leq 2\exp\left(-\frac{K}{3\sum_j \sigma_j}\right).
\end{aligned}$$

$\square$

## F.3 Proof of Proposition E.4

To prove the proposition for sufficiently large $T$, we focus on the regime where $N \geq 10R^2\rho^2/M^2 + 2$. We first use proof by contradiction to show the existence of $k_0 \leq 10R^2\rho^2/M^2$ such that $||\boldsymbol{r}_{k_0}||_2 < M/\rho$. Assume the contrary, we have $||\boldsymbol{r}_k||_2 \geq M/\rho$ for all $k \leq 10R^2\rho^2/M^2$. Recall we have proved earlier that (see inequality (42))

$$\left(\boldsymbol{x}^* - \boldsymbol{x}_k^{(\mathrm{B})}\right) \cdot \boldsymbol{r}_k \geq 0.6||\boldsymbol{r}_k||_2^2. \tag{67}$$

This assumption implies that $R \geq 0.6M/\rho$ and $\left\|\boldsymbol{x}^* - \boldsymbol{x}_k^{(\mathrm{B})}\right\|_2 \geq 0.6M/\rho$ for all $k \leq 10R^2\rho^2/M^2$.

We characterize the evolution of $x_k^{(\mathrm{B})}$. By Cauchy's inequality and inequality (67),

$$\begin{aligned}
\left(\boldsymbol{x}^* - \boldsymbol{x}_k^{(\mathrm{B})}\right) \cdot \left(\boldsymbol{x}_{k+1}^{(\mathrm{B})} - \boldsymbol{x}_k^{(\mathrm{B})}\right) &\geq \left(\boldsymbol{x}^* - \boldsymbol{x}_k^{(\mathrm{B})}\right) \cdot \boldsymbol{r}_k - \frac{M}{\rho T^{0.2}}\left\|\boldsymbol{x}^* - \boldsymbol{x}_k^{(\mathrm{B})}\right\|_2 \\
&\geq 0.6||\boldsymbol{r}_k||_2^2 - \frac{M}{\rho T^{0.2}}\left\|\boldsymbol{x}^* - \boldsymbol{x}_k^{(\mathrm{B})}\right\|_2.
\end{aligned}$$

Note that our assumed lower bound on $N$ implies a lower bound on $T$. Numerically, one can prove that $T^{0.2} \geq 20\rho R/M$. Hence, the above inequality implies that

$$\left(x^* - x_k^{(\mathrm{B})}\right) \cdot \left(x_{k+1}^{(\mathrm{B})} - x_k^{(\mathrm{B})}\right) \geq 0.6\|r_k\|_2^2 - \frac{0.05M^2}{\rho^2 R}\left\|x^* - x_k^{(\mathrm{B})}\right\|_2.$$

Then, by following the proof steps in Proposition E.1, we have that

$$\left\|x^* - x_{k+1}^{(\mathrm{B})}\right\|_2^2 - \left\|x^* - x_k^{(\mathrm{B})}\right\|_2^2 = -2\left(x^* - x_k^{(\mathrm{B})}\right) \cdot \left(x_{k+1}^{(\mathrm{B})} - x_k^{(\mathrm{B})}\right) + \left\|x_{k+1}^{(\mathrm{B})} - x_k^{(\mathrm{B})}\right\|_2^2$$

$$\leq -1.2\|r_k\|_2^2 + \frac{0.1M^2}{\rho^2 R}\left\|x^* - x_k^{(\mathrm{B})}\right\|_2 + \left(\frac{M}{\rho}\right)^2, \qquad (68)$$

where the second step is due to the construction of $x_{k+1}^{(\mathrm{B})}$ in Algorithm 4. Recall that $\left\|x^* - x_0^{(\mathrm{B})}\right\|_2 \leq R$. The above inequality implies that if $\|r_k\|_2 \geq M/\rho$ for all $k \leq 10R^2\rho^2/M^2$, then $\left\|x^* - x_k^{(\mathrm{B})}\right\|_2$ is non-increasing and reaches below 0 at $k = \lfloor 10R^2\rho^2/M^2 \rfloor + 1$. However, this contradicts the fact that $\|r_k\|_2$ is non-negative, and we must conclude the existence of $k_0 \leq 10R^2\rho^2/M^2$ such that $\|r_{k_0}\|_2 < M/\rho$.

Now consider any index $k$ with $\|r_k\|_2 < M/\rho$. By the construction of $r_k$, we have that $\nabla f(x_k) = -r_k \cdot \nabla^2 f(x_k)$. Then, by the Lipschitz Hessian condition,

$$\left\|\nabla f(x_{k+1}) - (x_{k+1} - x_k - r_k) \cdot \nabla^2 f(x_k)\right\|_2$$
$$= \left\|\nabla f(x_{k+1}) - \nabla f(x_k) - (x_{k+1} - x_k) \cdot \nabla^2 f(x_k)\right\|_2$$
$$\leq \frac{\rho}{2}\|x_{k+1} - x_k\|_2^2. \qquad (69)$$

Using the strong convexity assumption and triangle inequality, the above bound implies that

$$\left\|(\nabla^2 f(x_{k+1}))^{-1}\nabla f(x_{k+1})\right\|_2$$
$$\leq \left\|(x_{k+1} - x_k - r_k) \cdot \nabla^2 f(x_k) \cdot (\nabla^2 f(x_{k+1}))^{-1}\right\|_2 + \frac{\rho}{2M}\|x_{k+1} - x_k\|_2^2.$$

Note that the first term in the bound above is upper bounded by the product of $\|x_{k+1} - x_k - r_k\|_2$ and the spectral norm of $\nabla^2 f(x_k) \cdot (\nabla^2 f(x_{k+1}))^{-1}$. By the Lipschitz Hessian condition and strong convexity, this spectrum norm is further bounded by $1 + \frac{\rho}{M}\|x_{k+1} - x_k\|_2$. Therefore,

$$\left\|(\nabla^2 f(x_{k+1}))^{-1}\nabla f(x_{k+1})\right\|_2$$
$$\leq \|x_{k+1} - x_k - r_k\|_2 \cdot \left(1 + \frac{\rho}{M}\|x_{k+1} - x_k\|_2\right) + \frac{\rho}{2M}\|x_{k+1} - x_k\|_2^2. \qquad (70)$$

We use inequality (70) to bound $\|r_k\|_2$ recursively. Assume $T$ is sufficiently large such that $T^{0.2} \geq 20$. As a rough estimate, we have

$$\left\|(\nabla^2 f(x_{k+1}))^{-1}\nabla f(x_{k+1})\right\|_2 \leq \frac{M}{20\rho} \cdot 2 + \frac{M}{2\rho} \leq 0.6\frac{M}{\rho}.$$

Recall we can find $k_0 \leq 10R^2\rho^2/M^2$ such that $\|r_{k_0}\|_2 < M/\rho$. By induction, we have $\|r_k\|_2 \leq 0.6M/\rho$ for all $k > k_0$. Hence, when $k > k_0$, inequality (70) implies the following relation, where the RHS is obtained by triangle inequality and the definition of $E_{N-1} = 0$.

$$\|r_{k+1}\|_2 \leq \frac{M}{\rho T^{0.2}} \cdot \left(1 + \frac{\rho}{M}\|r_k\|_2 + \frac{1}{T^{0.2}}\right) + \frac{\rho}{2M}\left(\|r_k\|_2 + \frac{M}{\rho T^{0.2}}\right)^2.$$

Therefore, by induction, we have

$$\|r_k\|_2 \leq \frac{M}{\rho}\max\left\{\frac{0.6}{2^{2^{k-k_0-2}}}, \frac{2}{T^{0.2}}\right\}$$

for any $k > k_0 + 1$, and numerically, $\|r_{N-1}\|_2 \leq 2MT^{-0.2}/\rho$ if $T^{0.1} \geq 2k_0 + 6$.

## F.4 Proof of Proposition E.5

*Proof of inequality* (55). We prove the inequality by considering two possible cases. In the first case, we assume that the $\ell_2$ norms of both $H^{-1}\boldsymbol{m}$ and $H'^{-1}\boldsymbol{m}'$ are no greater than $R_0$. In this case, we have $H_{m^*} = H$ and $H'_{m'^*} = H'$. Hence,

$$
\begin{aligned}
H_{m^*}^{-1}\boldsymbol{m} - H_{m'^*}'^{-1}\boldsymbol{m}' &= H^{-1}\boldsymbol{m} - H'^{-1}\boldsymbol{m}' \\
&= H^{-1}\left((\boldsymbol{m} - \boldsymbol{m}') + (H' - H)H'^{-1}\boldsymbol{m}'\right).
\end{aligned}
\tag{71}
$$

By the fact that all eigenvalues of $H$ are lower bounded by $M$ and the triangle inequality,

$$
\begin{aligned}
\left\|H_{m^*}^{-1}\boldsymbol{m} - H_{m'^*}'^{-1}\boldsymbol{m}'\right\|_2 &\leq M^{-1}\left(\|\boldsymbol{m} - \boldsymbol{m}'\|_2 + \|H' - H\|_{\mathrm{F}}\|H'^{-1}\boldsymbol{m}'\|_2\right) \\
&\leq M^{-1}\left(\|\boldsymbol{m} - \boldsymbol{m}'\|_2 + \|H' - H\|_{\mathrm{F}} \cdot R_0\right).
\end{aligned}
\tag{72}
$$

Then, the needed inequality is obtained by $\left\|H_{m^*}^{-1}\boldsymbol{m} - H_{m'^*}'^{-1}\boldsymbol{m}'\right\|_2 \leq 2R_0$, which follows from the construction of $H_{m^*}$, $H'_{m'^*}$ and triangle inequality.

For the other case, we have $\max\left\{\|H^{-1}\boldsymbol{m}\|_2, \|H'^{-1}\boldsymbol{m}'\|_2\right\} > R_0$. Without loss of generality, we assume that $m^* \geq m'^*$. To be rigorous, here we adopted the convention that $m^* = -\infty$ if the $\ell_2$ norms of $H^{-1}\boldsymbol{m}$ is no greater than $R_0$, and the same for $m'^*$ accordingly. Based on this assumption, the condition in this case can be simplified as $\|H^{-1}\boldsymbol{m}\|_2 > R_0$, and we have that $\|H_{m^*}^{-1}\boldsymbol{m}\|_2 = R_0$. Furthermore, we also have $m^* > M$.

To prove the needed inequality, we introduce an intermediate variable $H'_{m^*}$, which is defined as the symmetric matrix sharing the eigenbasis of $H'$, but with each eigenvalue $\lambda$ replaced with $\max\{\lambda, m^*\}$. Note that $H_{m^*}$ and $H'_{m^*}$ are obtained by projecting $H$ and $H'$ to a convex set of matrices under the Frobenius norm. We have that

$$
\|H'_{m^*} - H_{m^*}\|_{\mathrm{F}} \leq \|H' - H\|_{\mathrm{F}}.
\tag{73}
$$

Therefore, by following the same steps in the first case and noting that all eigenvalues of $H'_{m^*}$ are lower bounded by $m^*$, we have that

$$
\left\|H_{m^*}^{-1}\boldsymbol{m} - H_{m^*}'^{-1}\boldsymbol{m}'\right\|_2 \leq m^{*-1}\left(\|\boldsymbol{m} - \boldsymbol{m}'\|_2 + \|H' - H\|_{\mathrm{F}} \cdot R_0\right).
$$

Compare the above to inequality (55), it remains to prove that

$$
\left\|H_{m^*}^{-1}\boldsymbol{m} - H_{m'^*}'^{-1}\boldsymbol{m}'\right\|_2^2 \leq \frac{2R_0 m^*}{M} \cdot \left\|H_{m^*}^{-1}\boldsymbol{m} - H_{m^*}'^{-1}\boldsymbol{m}'\right\|_2.
\tag{74}
$$

For brevity, we denote that

$$
\begin{aligned}
\boldsymbol{a} &\triangleq H_{m^*}^{-1}\boldsymbol{m}, \\
\boldsymbol{b} &\triangleq H_{m^*}'^{-1}\boldsymbol{m}', \\
\boldsymbol{c} &\triangleq H_{m'^*}'^{-1}\boldsymbol{m}', \\
\alpha &\triangleq M/m^*.
\end{aligned}
$$

In the eigenbasis of $H'$, it is clear that

$$
\|\boldsymbol{b} - \alpha\boldsymbol{c}\|_2 \leq (1 - \alpha)\|\boldsymbol{c}\|_2.
$$

Hence, by Cauchy's inequality,

$$
\boldsymbol{a} \cdot (\boldsymbol{b} - \alpha\boldsymbol{c}) \leq \|\boldsymbol{a}\|_2 \cdot \|\boldsymbol{b} - \alpha\boldsymbol{c}\|_2 \leq (1 - \alpha)\|\boldsymbol{a}\|_2 \cdot \|\boldsymbol{c}\|_2.
\tag{75}
$$

Recall that $\|\boldsymbol{c}\|_2 \leq R_0$ and in this case we have $\|\boldsymbol{a}\|_2 = R_0$. Therefore, the RHS of the above inequality is upper bounded by $(1 - \alpha)\|\boldsymbol{a}\|_2^2$, and we have

$$
\boldsymbol{a} \cdot (\boldsymbol{a} - \boldsymbol{c}) \leq \frac{1}{\alpha}\boldsymbol{a} \cdot (\boldsymbol{a} - \boldsymbol{b}) \leq \frac{1}{\alpha}R_0\|\boldsymbol{a} - \boldsymbol{b}\|_2,
$$

where the first step above is equivalent to inequality (75), and the second step is due to Cauchy's inequality. Finally, it remains to notice that the LHS of inequality (74) equals $\|\boldsymbol{a} - \boldsymbol{c}\|_2^2$, which is upper bounded by the LHS of the above inequality, and its RHS equals the RHS of the above inequality. Hence, inequality (74) is proved. $\quad\square$

*Proof of inequality* (56). Firstly, if $m^* = m'^*$, we follow similar arguments from equation (71) to inequality (72). I.e., in this case, we have

$$H'_{m'^*}\left(H_{m^*}^{-1}\boldsymbol{m} - H'^{-1}_{m'^*}\boldsymbol{m}'\right) = (\boldsymbol{m} - \boldsymbol{m}') + (H'_{m'^*} - H_{m^*})H_{m^*}^{-1}\boldsymbol{m}. \tag{76}$$

Hence, by triangle inequality and inequality (73),

$$\left|\left|H'_{m'^*}\left(H_{m^*}^{-1}\boldsymbol{m} - H'^{-1}_{m'^*}\boldsymbol{m}'\right)\right|\right|_2 \leq ||\boldsymbol{m} - \boldsymbol{m}'||_2 + ||H'_{m'^*} - H_{m^*}||_{\mathrm{F}}\left|\left|H_{m^*}^{-1}\boldsymbol{m}\right|\right|_2$$
$$\leq ||\boldsymbol{m} - \boldsymbol{m}'||_2 + ||H' - H||_{\mathrm{F}} \cdot R_0. \tag{77}$$

Then, for $m^* > m'^*$, we define $H'_{m^*}$ and $\boldsymbol{a}$, $\boldsymbol{b}$, $\boldsymbol{c}$ as in the earlier proof steps. We first prove the following key inequality.

$$||\boldsymbol{a} - \boldsymbol{c}||_2 \cdot ||\boldsymbol{b} - \boldsymbol{c}||_2 \leq 2||\boldsymbol{a} - \boldsymbol{b}||_2 \cdot ||\boldsymbol{a}||_2. \tag{78}$$

Recall the assumption in this case implies that $||\boldsymbol{a}||_2 = R_0$. By taking the squares on both sides, the inequality above is equivalent to the following linear inequality of vector $\boldsymbol{a}$.

$$\boldsymbol{a} \cdot \left(8R_0^2\,\boldsymbol{b} - 2||\boldsymbol{b} - \boldsymbol{c}||_2^2\,\boldsymbol{c}\right) \leq 4R_0^2 \cdot \left(R_0^2 + ||\boldsymbol{b}||_2^2\right) - ||\boldsymbol{b} - \boldsymbol{c}||_2^2 \cdot \left(R_0^2 + ||\boldsymbol{c}||_2^2\right). \tag{79}$$

By Cauchy's inequality, the LHS of inequality (79) is upper bounded by $||\boldsymbol{a}||_2 \cdot ||8R_0^2\,\boldsymbol{b} - 2||\boldsymbol{b} - \boldsymbol{c}||_2^2\,\boldsymbol{c}||_2$. The coefficient of $||\boldsymbol{a}||_2$ in this expression can be further characterized as follows.

$$\left|\left|8R_0^2\,\boldsymbol{b} - 2||\boldsymbol{b} - \boldsymbol{c}||_2^2\,\boldsymbol{c}\right|\right|_2^2 = 4 \cdot \left(4R_0^2 - ||\boldsymbol{b} - \boldsymbol{c}||_2^2\right)\left(4R_0^2||\boldsymbol{b}||_2^2 - ||\boldsymbol{b} - \boldsymbol{c}||_2^2||\boldsymbol{c}||_2^2\right)$$
$$+ 16R_0^2 \cdot ||\boldsymbol{b} - \boldsymbol{c}||_2^4$$
$$= \frac{1}{R_0^2}\left(4R_0^2 \cdot \left(R_0^2 + ||\boldsymbol{b}||_2^2\right) - ||\boldsymbol{b} - \boldsymbol{c}||_2^2 \cdot \left(R_0^2 + ||\boldsymbol{c}||_2^2\right)\right)^2$$
$$- \frac{1}{R_0^2}\left(4R_0^2 \cdot \left(R_0^2 - ||\boldsymbol{b}||_2^2\right) - ||\boldsymbol{b} - \boldsymbol{c}||_2^2 \cdot \left(R_0^2 - ||\boldsymbol{c}||_2^2\right)\right)^2$$
$$+ 16R_0^2 \cdot ||\boldsymbol{b} - \boldsymbol{c}||_2^4. \tag{80}$$

We prove that the contribution from the second term and the third term in the above expression is non-positive. To that end, note that the definition of $H'_{m^*}$, $H'_{m'^*}$ and the assumption of $m^* > m'^*$ imply that $(\boldsymbol{c} - \boldsymbol{b}) \cdot \boldsymbol{b} \geq 0$. We have the following inequalities.

$$||\boldsymbol{b} - \boldsymbol{c}||_2^2 + ||\boldsymbol{b}||_2^2 \leq ||\boldsymbol{c}||_2^2 \leq R_0^2. \tag{81}$$

Therefore, $0 \leq 4R_0^2 \cdot \left(R_0^2 - ||\boldsymbol{b}||_2^2\right) - ||\boldsymbol{b} - \boldsymbol{c}||_2^2 \cdot \left(R_0^2 - ||\boldsymbol{c}||_2^2\right) \leq 4R_0^2 \cdot ||\boldsymbol{b} - \boldsymbol{c}||_2^2$, and equation (80) implies that

$$\boldsymbol{a} \cdot \left(8R_0^2\,\boldsymbol{b} - 2||\boldsymbol{b} - \boldsymbol{c}||_2^2\,\boldsymbol{c}\right) \leq \left|4R_0^2 \cdot \left(R_0^2 + ||\boldsymbol{b}||_2^2\right) - ||\boldsymbol{b} - \boldsymbol{c}||_2^2 \cdot \left(R_0^2 + ||\boldsymbol{c}||_2^2\right)\right|.$$

By utilizing the above bound, inequality (79) is proved by noting that its RHS is non-negative, which can be proved using inequality (81). As mentioned earlier, this implies inequality (78).

To proceed further, we note that $\boldsymbol{b} - \boldsymbol{c}$ lies in the eigenspace of $H'_{m^*}$ associated with eigenvalue $m^*$. Hence,

$$H'_{m^*}\left(\boldsymbol{a} - \boldsymbol{c}\right) = H'_{m^*}\left(\boldsymbol{a} - \boldsymbol{b}\right) + m^*\left(\boldsymbol{b} - \boldsymbol{c}\right).$$

Therefore, by triangle inequality, we have

$$\left|\left|H'_{m^*}\left(H_{m^*}^{-1}\boldsymbol{m} - H'^{-1}_{m'^*}\boldsymbol{m}'\right)\right|\right|_2 \leq ||H'_{m^*}\left(\boldsymbol{a} - \boldsymbol{b}\right)||_2 + m^*\,||\boldsymbol{b} - \boldsymbol{c}||_2. \tag{82}$$

Note that by inequality (78) and the fact that all eigenvalues of $H'_{m^*}$ are lower bounded by $m^*$, we have

$$m^*\,||\boldsymbol{b} - \boldsymbol{c}||_2 \leq \frac{2R_0}{||\boldsymbol{a} - \boldsymbol{c}||_2}\,||H'_{m^*}\left(\boldsymbol{a} - \boldsymbol{b}\right)||_2.$$

Therefore, it remains to upper bound the $\ell_2$ norm of $H'_{m^*}\left(\boldsymbol{a} - \boldsymbol{b}\right)$.

By the definition of vectors $\boldsymbol{a}$, $\boldsymbol{b}$,

$$H'_{m^*}\left(\boldsymbol{a} - \boldsymbol{b}\right) = (\boldsymbol{m} - \boldsymbol{m}') + (H'_{m^*} - H_{m^*})\,\boldsymbol{a}. \tag{83}$$

The triangle inequality implies that

$$\left\|H'_{m^*}\left(\boldsymbol{a}-\boldsymbol{b}\right)\right\|_2 \le \left\|\boldsymbol{m}-\boldsymbol{m}'\right\|_2 + R_0 \left\|H'_{m^*}-H_{m^*}\right\|_{\mathrm{F}}. \tag{84}$$

Hence,

$$\begin{aligned}
\left\|H'_{m^*}\left(H_{m^*}^{-1}\boldsymbol{m} - H'^{-1}_{m'^*}\boldsymbol{m}'\right)\right\|_2 &\le \left(1 + \frac{2R_0}{\left\|\boldsymbol{a}-\boldsymbol{c}\right\|_2}\right) \\
&\quad \cdot \left(\left\|\boldsymbol{m}-\boldsymbol{m}'\right\|_2 + R_0 \left\|H'_{m^*}-H_{m^*}\right\|_{\mathrm{F}}\right) \\
&\le \left(1 + \frac{2R_0}{\left\|H_{m^*}^{-1}\boldsymbol{m} - H'^{-1}_{m'^*}\boldsymbol{m}'\right\|_2}\right) \\
&\quad \cdot \left(\left\|\boldsymbol{m}-\boldsymbol{m}'\right\|_2 + R_0 \left\|H-H'\right\|_{\mathrm{F}}\right),
\end{aligned}$$

where the last step is due to inequality (73). Thus, inequality (56) is implied by the semi-positive-definiteness of $H'_{m^*} - H'_{m'^*}$.

Finally, when $m^* < m'^*$, we let $H_{m'^*}$ denote the symmetric matrix sharing the same eigenbasis of $H$, but with each eigenvalue $\lambda$ replaced by $\max\{\lambda, m'^*\}$. Due to the equivalence of $H$ and $H'$, our earlier proof steps imply that

$$\begin{aligned}
\left\|H_{m^*}\left(H_{m^*}^{-1}\boldsymbol{m} - H'^{-1}_{m'^*}\boldsymbol{m}'\right)\right\|_2 &\le \left(1 + \frac{2R_0}{\left\|H_{m^*}^{-1}\boldsymbol{m} - H'^{-1}_{m'^*}\boldsymbol{m}'\right\|_2}\right) \\
&\quad \cdot \left(\left\|\boldsymbol{m}-\boldsymbol{m}'\right\|_2 + R_0 \left\|H-H'\right\|_{\mathrm{F}}\right).
\end{aligned}$$

Hence, by triangle inequality, we can use the above bound as follows.

$$\begin{aligned}
\left\|H'_{m'^*}\left(H_{m^*}^{-1}\boldsymbol{m} - H'^{-1}_{m'^*}\boldsymbol{m}'\right)\right\|_2 &\le \left\|\left(H_{m'^*} - H'_{m'^*}\right)\left(H_{m^*}^{-1}\boldsymbol{m} - H'^{-1}_{m'^*}\boldsymbol{m}'\right)\right\|_2 \\
&\quad + \left\|H_{m'^*}\left(H_{m^*}^{-1}\boldsymbol{m} - H'^{-1}_{m'^*}\boldsymbol{m}'\right)\right\|_2 \\
&\le \left\|H_{m'^*} - H'_{m'^*}\right\|_{\mathrm{F}} \left\|H_{m^*}^{-1}\boldsymbol{m} - H'^{-1}_{m'^*}\boldsymbol{m}'\right\|_2 \\
&\quad + \left\|H_{m'^*}\left(H_{m^*}^{-1}\boldsymbol{m} - H'^{-1}_{m'^*}\boldsymbol{m}'\right)\right\|_2.
\end{aligned}$$

Note that $\left\|H_{m^*}^{-1}\boldsymbol{m} - H'^{-1}_{m'^*}\boldsymbol{m}'\right\|_2 \le 2R_0$. By inequality (73), it is clear that

$$\begin{aligned}
\left\|H'_{m'^*}\left(H_{m^*}^{-1}\boldsymbol{m} - H'^{-1}_{m'^*}\boldsymbol{m}'\right)\right\|_2 &\le \left(3 + \frac{2R_0}{\left\|H_{m^*}^{-1}\boldsymbol{m} - H'^{-1}_{m'^*}\boldsymbol{m}'\right\|_2}\right) \\
&\quad \cdot \left(\left\|\boldsymbol{m}-\boldsymbol{m}'\right\|_2 + R_0 \left\|H-H'\right\|_{\mathrm{F}}\right).
\end{aligned}$$

$\square$

