# OpenReview forum: "Stochastic Zeroth-Order Optimization under Strongly Convexity and Lipschitz Hessian: Minimax Sample Complexity"
_NeurIPS.cc/2024/Conference — NeurIPS 2024 poster_

### Official Review · Reviewer_cqSV · 2024-06-25

**Soundness:** 2
**Presentation:** 1
**Contribution:** 2
**Rating:** 5
**Confidence:** 3

**Summary:**

The paper consider the problem of optimizing second-order smooth and strongly convex functions where the algorithm is only accessible to noisy evaluations of the objective function it queries. Authors provide the first tight characterization for the rate of the minimax simple regret by developing matching upper and lower bounds. They propose an algorithm that features a combination of a bootstrapping stage and a mirror-descent stage.

**Strengths:**

The authors consider a problem formulation where the gradient of the function or higher order derivatives are not available. This problem formulation is of extreme interest in the field of Machine Learning. However, the work has many weaknesses (see below).

**Weaknesses:**

1. Almost no mention is made of work motivation. This is an important part that **cannot be missed**...




2. Also, this paper gives the impression that **it was produced clearly not in this year**, since the main references are **from 2021** and earlier. But since 2021, the research, in particular in the area of zero-order optimization has a lot of interest in the community and is highly advanced compared to what the authors cited as related works. For example, in addition to a number of cited papers studying the zero-order optimization problem with the assumption of increased smoothness of the function, there is already a more recent paper that has been missed: _Akhavan et al. (2023)_ [1], which proposes an improved analysis of gradient approximation bias estimation as well as second moment estimation. Similarly, there is a weakly described section presenting different gradient approximations, where the main focus fell on the smoothing vector (Gaussian vector or Randomized vector). But, there are already a number of papers, such as _Gasnikov et al. (2024)_ [2], which have shown that the randomized approximation clearly performs better on practical experiments than the Gaussian approximation. It seems better to give an overview of different gradient approximations, including $l_1$ randomization, in this section


3. Since I have already mentioned the topic of practical experiments, it is important to note that despite the fact that the authors provide mainly theoretical work, a paper that is intended for presentation at the NeurIPS conference should **show the effectiveness of the proposed results** in real life. This paper lacks any experiments, which is a major weakness of the paper.

4. The structure of the article is poorly chosen... It seems that the sections "Main contribution", "Discussion" and "Conclusion" would **have improved the presentation** of the article's results.

5. Regarding the results themselves, essentially **only one result is presented**, which is a minor contribution. It would be interesting to consider when the function is not only twice differentiable but also has a higher order of smoothness, as well as other settings of the problem: convex, non-convex, maybe the Polyak–Lojasiewicz condition, etc.

[1] Akhavan A. et al. (2023). Gradient-free optimization of highly smooth functions: improved analysis and a new algorithm //arXiv preprint arXiv:2306.02159.

[2] Gasnikov, A. et al. (2024). Highly smooth zeroth-order methods for solving optimization problems under the PL condition. Computational Mathematics and Mathematical Physics.

**Questions:**

I have a few questions and suggestions:

- It seems that Theorem 3.1 can be improved in terms of dependence on the problem dimension $d$ using tricks from the work of _Akhavan et al. (2023)_ [1]

- Can the authors explain what the final result would look like if $w_t$ does not necessarily have zero mean, but is independent of vector $e$ (randomization on the sphere)? Also, if we consider deterministic noise $|\delta(x)| \leq \Delta$ as noise.

[1] Akhavan A. et al. (2023). Gradient-free optimization of highly smooth functions: improved analysis and a new algorithm //arXiv preprint arXiv:2306.02159.

---

> ### Author Rebuttal · Authors · 2024-08-07
>
> Thank you for your feedback. Please find our responses below.
>
> > Almost no mention is made of work motivation.
>
> - The importance of stochastic optimization is clearly mentioned in our introduction.
>
> - The motivation for this work stems from a lack of understanding of the sample complexity under convexity and higher-order smoothness. As stated in Lines 39-46 and summarized in Table 1, the problem of finding tighter complexity bounds for higher-order smooth functions has been widely studied, but the existing sample complexity bounds all have an apparent gap in the regime of $k \geq 2$.
>
> > Also, this paper gives the impression that it was produced clearly not in this year, since the main references are from 2021 and earlier.
>
> - We have cited papers from beyond 2021, such as Lattimore & György (2023) and Yu et al. (2023).
>
> - Thank you for your suggestion on references. However, the references you mentioned, Akhavan et al. (2023) and Gasnikov et al. (2024), address different research questions from those in our paper. They focus on zeroth-order optimization under assumptions such as the PL condition and higher-order Hölder smoothness. In contrast, we assume strong convexity and a Lipschitz Hessian. More specifically, similar to other related works, Akhavan et al. (2023) adopt the additional assumption of the Lipschitz gradient while we do not impose this condition. Their $O(r^2)$ bound on the bias of the $L_2$ gradient estimator for $\beta=3$ is implied and strengthened by our bound in Theorem 4.1 as we discussed in Appendix A. On the other hand, Gasnikov et al. (2024) focus on adversarial bandit problems with (almost) adversarial noises, whereas we consider stochastic bandit problems with i.i.d. random noises.
>
>
> > … a paper that is intended for presentation at the NeurIPS conference should show the effectiveness of the proposed results in real life. This paper lacks any experiments, which is a major weakness of the paper.
>
> - Our work addresses the theoretical question of bandit optimization under strong convexity and Lipschitz Hessian. We provide matching lower and upper bounds for simple regret, thereby fully resolving the complexity of this fundamental problem. Given the theoretical nature of our contributions, experimental validation is outside the scope of this study.
>
> > It seems that Theorem 3.1 can be improved in terms of dependence on the problem dimension d using tricks from the work of Akhavan et al. (2023)
>
> - Under the current problem setting, the tricks from Akhavan et al. (2023) cannot improve the dependence on $d$ in Theorem 3.1. This is because our lower bound in Theorem 3.2 already matches Theorem 3.1 in its dependence on $d$.
> Furthermore, the dependence on $d$ in Akhavan et al. (2023) is suboptimal compared to our bound. Specifically, their rate is $d^{2 - \frac{2}{\beta}}$ (where $\beta \geq 1$ is the order of Hölder smoothness), while our rate is linear in $d$. Therefore, we believe that applying techniques from Akhavan et al. (2023) is unlikely to improve the dependence on $d$.
>
> > Can the authors explain what the final result would look like if 𝑤𝑡 does not necessarily have zero mean, but is independent of vector 𝑒 (randomization on the sphere)? Also, if we consider deterministic noise |𝛿(𝑥)|≤Δ as noise.
>
> - Thank you for pointing out this interesting future direction. In our work, we consider the stochastic bandit setting where the noises are i.i.d. random variables. If $w_t$’s are dependent on $e$ or are deterministic, the problem would fall into the category of adversarial bandits, which requires fundamentally different approaches and is outside the scope of our paper.

---

> > ### Comment · Reviewer_cqSV · 2024-08-08
> >
> > Dear Authors,
> >
> > I thank you for your replies. However, the authors have confirmed my concerns (ignoring some of the weaknesses in the rebuttals)... I will discuss the readiness of the article for presentation at the conference with other Reviewers and Area Chair before making a final decision. At the moment, I maintain a score of 2.

---

> > > ### Author Response · Authors · 2024-08-09
> > >
> > > Dear reviewer cqSV,
> > >
> > > Thank you for your recent update.
> > >
> > > We noticed that you mentioned, “the authors have confirmed my concerns.” We would appreciate your clarification on how our response confirmed your concern that "Theorem 3.1 can be improved in terms of dependence on the problem dimension using ..."
> > >
> > > As we have explained in our rebuttal, "our lower bound in Theorem 3.2 already matches Theorem 3.1 in terms of dependence on the problem dimension." making such improvements mathematically infeasible.
> > >
> > > Could you provide further details on how you believe the dependence on d could be improved?
> > >
> > > Best,
> > > Authors of Submission4474

---

> > > > ### Comment · Reviewer_cqSV · 2024-08-09
> > > >
> > > > Dear Authors,
> > > >
> > > > After discussing the readiness of the paper with the other Reviewers and Area Chair, I raise my score to 5. However, I recommend that the authors take into account all the comments and significantly improve the paper, as it is currently at least borderline...
> > > >
> > > > Good luck!

---

### Official Review · Reviewer_X3iq · 2024-07-09

**Soundness:** 3
**Presentation:** 3
**Contribution:** 3
**Rating:** 6
**Confidence:** 3

**Summary:**

The paper studies zero order stochastic optimization (the learner has access to noisy function evaluation only) assuming the objective is $M$-strongly convex and has a $\rho$-Lipschitz (in forbenius norm) Hessian. Matching upper and lower bounds are presented, which establish a tight result of suboptimality $\Theta(d T^{-2/3}M^{-1})$ (w.h.p.) after $T$ function evaluations.

**Strengths:**

* A tight result is achieved for the setting studied.
* On the technical side, the gradient evaluation procedure in the final stage of the algorithm uses samples from the ellipsoid induced by the Hessian approximation rather than opting for isotropic sampling. Further, a novel approach for analyzing the bias and variance of this estimation procedure is presented that yields sharp bounds. This technique has not been used by prior works, and seems crucial for the minimax optimal result.
* This result improves over the most closely related prior state-of-the-art Novitskii & Gasnikov (2021) by a factor of $d^{1/3}$, who require an additional Lipschitz gradient assumption (see discussion in Appendix A).

**Weaknesses:**

Nothing in particular.

**Questions:**

- Lines 125-126: 1-subgaussianity already implies variance $\leq 1$.
- Akhavan et al. (2020) consider the Lipschitz gradient setting and do not require a Lipschitz Hessian. Hence it seems their results and yours are incomparable (correct?). Given this I did not understand lines 315-317: "On the other hand ... well." Do you include Akhavan et al in this statement? It doesn't seem true that a direct application of your algorithm order wise improves the result of Akhavan et al because your result does not apply in their setting.
- Since we consider a twice differentiable objective over a bounded domain, it follows the gradient is in fact Lipschitz. With this in mind, are the assumptions of Novitskii & Gasnikov (2021) indeed stronger than those made in this work?

---

> ### Author Rebuttal · Authors · 2024-08-06
>
> We appreciate the constructive feedback from the reviewer. Below are the point-to-point responses to the comments.
>
> >Implication of bounded variance from 1-subgaussianity
>
> We would like to clarify that our intention in mentioning both assumptions was to use the sub-Gaussian assumption to simplify the algorithmic design and meanwhile explicitly state the bounded variance condition for readers who may not be familiar with the properties of sub-Gaussian variables.
>
> However, we want to emphasize that, of the two assumptions, the bounded variance condition is more fundamental to our results. The sub-Gaussian assumption can be removed by employing more sophisticated mean-estimation methods, such as median-of-means instead of naive averaging. In the revised manuscript, we will clarify that the use of the stronger sub-Gaussian assumption is for simplicity. We will also refer to references related to these alternative mean estimators, e.g., [1].
>
>
> [1] A. Nemirovskii and D. Yuom. Problem Complexity and Method Efficiency in Optimization,". Wiley, 1983.
>
> > Applicability of our result to Akhavan et al. (2020) and the comparison in line 315-317
>
> As mentioned in Appendix A of our paper, although Akhavan et al. (2020) did not directly assume a Lipschitz Hessian, our result can be related to their $\beta=3$ smoothness condition (which we refer to as the $k=3$ smoothness condition in our paper). The bound we discuss in lines 315-317 corresponds to Lemma 2.3 in Akhavan et al. (2020). As a sanity check, the prior work needed to assume this higher-order smoothness condition to obtain a bound that scales with $r^2$ for the gradient estimator. This is because, with only the Lipschitz gradient condition, the best achievable bound scales with $r$, which is worse by a factor of $1/r$.
>
>
> > Is Lipschitz gradient over bound domain in Novitskii & Gasnikov (2021) indeed a stronger assumption?
>
> We would like to clarify that although the twice differentiability implies the finiteness of the second-order derivatives over a compact set, it does not imply that the gradient is Lipschitz with a fixed constant such as $\rho$ or $L$ even for a fixed compact set. In other words, the Hessian can be finite but unbounded. A simple example is that the function class of all quadratic polynomials (with arbitrarily large second-order derivatives) satisfies  Lipschitz Hessian but not Lipschitz Gradient. Therefore, the Lipschitz gradient condition has to be explicitly assumed in prior works in addition to the higher-order smoothness conditions.
>
> Notably, even in the classical analysis of Newton's method, which assumes zero-error observations, an additional constant factor associated with the Lipschitz gradient condition is used to obtain non-trivial complexity bounds (e.g., see [2], Section 9.5.3). This demonstrates the difficulty of removing this smoothness condition. We introduced a non-trivial modification of the Newton step in the bootstrapping stage to achieve a uniform convergence guarantee of $E[f(x_B)-f^*]=O(1)$ (see Algorithm 4), which removes the need for the Lipschitz gradient condition. We consider this to be one of our main technical contributions.
>
> [2] Stephen P Boyd and Lieven Vandenberghe. Convex optimization. Cambridge university press, 2004.

---

### Official Review · Reviewer_QNuA · 2024-07-12

**Soundness:** 3
**Presentation:** 2
**Contribution:** 3
**Rating:** 5
**Confidence:** 3

**Summary:**

This work studies the convergence of zeroth order stochastic optimization for a class of strongly convex, second-order smooth objective functions.  The authors assume that the noisy one-point feedback oracle is available, and the additive noise is subgaussian. Both the asymptotic upper bound and the matching lower bound for the minimax simple regret are provided.

**Strengths:**

The authors propose a novel algorithm that combines gradient estimation, bootstrapping procedures, and a mirror descent procedure. This combination enables accurate estimation of gradients and Hessians under second-order smoothness, thereby achieving the optimal rate.

**Weaknesses:**

The idea of this work is interesting, but the presentation needs improvement. Some main results should be explicitly and rigorously stated as theorems, for example, eq (6). Additionally, it is unclear why the two stages in Algorithm 4 are necessary. Providing intuition and explanation for the two stages and the choice of parameters in Algorithm 4 would be beneficial.

Furthermore, the proof of the upper bounds contains gaps. The result obtained in line 601 is $\lim_{T\to\infty} T^{2/3}E[\|\nabla f(x_N^B)\|^3]=0$. It is unclear how $M$ and $\rho$ enter the final result and match eq(6) with the specific choice of $\epsilon$ in line 170. More detailed steps in the proof would help clarify this connection.

**Questions:**

1. Remark 4.2: I cannot understand the first sentence. Are there any established results for the case of the cubic polynomial function $f$?
2. Algorithm 4: Is the $\hat H$  returned by HessianEst invertible?
3. Proof of Theorem 3.1:
1)  In line 200, "First, recall our construction ensures that," could the author provide the proof of this statement? 2) In lines 203-204, the two displays omit the dimension dependence. Will this omission impact the final dimension dependence?
4. Line 358: I cannot see how to apply Lemma B.2, also known as Proposition 7 in [Yu et. al. 2023]. What is the $Var_{f\sim p}[f(x)]$?

**Limitations:**

The paper proposes a new algorithm but does not provide any numerical results to validate it. There are many interesting questions that could be investigated numerically regarding this method, such as its dependence on the length of the first and second stages.

---

> ### Author Rebuttal · Authors · 2024-08-06
>
> We appreciate the constructive feedback from the reviewer and will state equation (6) formally as a Theorem. Below are the point-to-point responses to the comments.
>
> >Necessity of the two-stage algorithm
>
>   The two phases of our algorithm are designed to handle distinct challenges. The first stage of our algorithm deals with potential initial conditions with unbounded gradients and highly skewed Hessians. To that end, we need a gradient estimator whose variance does not depend on the gradient norm to ensure the uniform boundedness of $E[f(x_B)-f^*]$. Therefore, we employ the coordinate-wise estimator in BootstrappingEst for gradient estimation in the first stage.
>
> In fact, the bootstrapping stage requires a non-trivial modification of the Newton step to achieve the uniform convergence guarantee of $E[f(x_B)-f^*]=O(1)$ (see Algorithm 4) without relying on the Lipschitz gradient condition assumed in prior works. We want to emphasize that even in the classical analysis of Newton's method, which assumes zero-error observations, the additional Lipschitz gradient condition is used to obtain non-trivial complexity bounds (e.g., see [1], Section 9.5.3). This highlights the difficulty of removing this smoothness condition, and we view the removal of this assumption in our work as one of the main technical contributions.
>
> [1] Stephen P Boyd and Lieven Vandenberghe. Convex optimization. Cambridge university press, 2004.
>
> The second stage of our algorithm is needed to finetune the result from the first stage to achieve the optimal sample complexity. Once the algorithm reaches a point sufficiently close to the global minimum, as specified in equation (6), the hyperellipsoid estimator in GradientEst provides a better bias-variance tradeoff (by a factor of $d$) as the third term on the RHS of inequality (3) becomes dominant. Using this alternative estimator in the second stage allows us to reduce the estimation error by a factor of $d^{2/3}$.
>
> > gap from line 601 to equation (6)
>
> Thanks for pointing this out. This step is due to the strong convexity condition, which implies that $\frac{||\nabla f(x)||_2^2}{2M} \geq f(x)-f^*$ for any $x\in\mathbb{R}^d$. Recall that in our main theorems we are considering the limits as $T\rightarrow +\infty$ for fixed $d$, $M$, $ \rho$, and $R$ (this will also be stated in the updated version of the theorem for equation (6)). The convergence in inequality (6) is obtained by simply plugging in the definition of $\epsilon$ and the inequality above obtained from strong convexity into the limit of $ 𝑇^{2/3} 𝐸[||∇𝑓(𝑥_𝑁^{(𝐵)})||_2^3]$.
>
> >The first sharpness statement in Remark 4.2?
>
> This claim states that for any fixed $\lambda_z$, $\rho$, and $d$, there exists a function $f$ in $\mathcal{F}(M,\rho,R)$ and a matrix $Z=\lambda_z I$ such that the bias of GradientEst is exactly given by inequality (2). One such example of $f$ can be constructed as a function that is locally a polynomial of degree $3$, with its cubit term given by $f(\boldsymbol{x} )=\frac{\rho}{6}\sum_{k}(x_k^3)$, where $x_k$ are the individual coordinates of $\boldsymbol{x}$. While this construction of $f$ is elementary, the bias of GradientEst over this function is derived from integration over the hypersphere, which is standard and well-established.
>
> >Is the $\hat{H}$ returned by HessianEst invertible?
>
> Note that at the end of this subroutine, we raised all eigenvalues of \hat{H} to at least M. Hence, the returned $\hat{H}$ is positive definite, which must be invertible.
>
> >Proof of Theorem 3.1: In line 200?
>
> The condition $||x_T-x_B||_2\leq M/\rho$ is guaranteed by the clipping step in the final stage in Algorithm 4, where we project $r$ to the $L_2$ ball of radius $\frac{M}{\rho}$, and $x_T$ is created with $x_B+r$.
>
> >In lines 203-204
>
> Similar to what we have clarified in the comment above for equation (6), we shall clarify that the notation $f=o(g)$ means that  $\lim_{T\rightarrow +\infty} f/g=0$ for any fixed $M$, $\rho$, $R$, and $d$. This property is independent of any multiplicative factor of $d$, and it does not impact the asymptotic result for $T$ being sufficiently large.
>
> > Line 358: how to apply Lemma B.2, and what is the 𝑉𝑎𝑟𝑓∼𝑝[𝑓(𝑥)]?
>
> We choose $p$ to be the uniform distribution over the function class $\{f_1,f_2\}$, as stated in line 347, so that each $f_k$ has a probability of $½$. To apply Lemma B.2, we verify that the condition of uniform sampling error in Lemma B.2 is satisfied, which is shown in lines 361-368. The $\textup{𝑉𝑎𝑟}_{f\sim p}[𝑓(x)]$ under this prior is simply $(\frac{f_1(x)-f_2(x)}{2})^2$, and it is upper bounded by $\pi^2 y_0^2$ as mentioned in line 346-347.

---

> > ### Comment · Reviewer_QNuA · 2024-08-11
> >
> > Thanks for the authors' response. After thoroughly reviewing it, I will maintain my current rating.

---

### Official Review · Reviewer_vYTV · 2024-07-12

**Soundness:** 2
**Presentation:** 2
**Contribution:** 3
**Rating:** 6
**Confidence:** 3

**Summary:**

The paper studies the problem of zero-order optimization of a strongly convex function whose Hessian is Lipschitz continuous. The proposed algorithm exploits zero-order information from the oracle to estimate the Hessian and gradient of the function at each iteration. Using these estimates, the authors employ a second-order method to achieve convergence for the optimization error. The results are asymptotic and valid for a sufficiently large number of function evaluations. The main contribution of the paper is improving the dependency of the achieved optimization error with respect to the dimension. The derived upper bound is minimax optimal with respect to the number of function evaluations, dimension, and the strong convexity parameter.

**Strengths:**

The result of the paper regarding the upper bound is a significant and surprising contribution. In the literature, there was an optimality gap with respect to the dimension: the existing lower bound was of the order $d$, while the upper bound was of the order $d^{4/3}$. The main anticipation was that the lower bound was not tight enough. However, the result of this paper shows that $d$ is minimax optimal, which I find to be a very valuable observation.

**Weaknesses:**

The main issue I encountered while reading the paper is that the authors should provide more motivation for their algorithm. For instance, they use two different gradient estimators, GradientEst and BootstrappingEst, but it is unclear why both are necessary and why one is not sufficient. The assumption on the noise is also more restrictive compared to previous work in the literature, see e.g. [1]. The authors assume that the noise is sub-Gaussian, whereas other papers have only assumed a finite second moment. The authors did not explain why this stronger restriction on the noise is needed.

The main algorithm of the paper, Algorithm 4, consists of two stages. While the first stage of the algorithm seems natural, the second stage is not well motivated.


[1] Arya Akhavan, Massimiliano Pontil, and Alexandre Tsybakov. Exploiting higher order smoothness in derivative-free optimization and continuous bandits. Advances in Neural Information Processing Systems, 2020.

**Questions:**

1. As I mentioned earlier, could the authors explain why there is a need for two different gradient estimators?

2. Why does the noise need to be sub-Gaussian? Specifically, the results of Theorem 4.1 also hold for any noise with a finite second moment.

3. Could the authors motivate the second stage of the algorithm? The authors mentioned, "The role of the final stage is to ensure that $f(x_B) - f(x^*)$ is sufficiently small with high probability," but why you need such a guarantee with high probability, and not just in expectation?

4. Could the authors explain why they need a high probability bound in Theorem 4.3 and why expectation bounds are not sufficient?

5. I am a bit confused about where the authors used Assumption A (3). The parameter $R$ does not appear anywhere in the algorithms or in the final bound.

6. It would be helpful if the authors provided a discussion on the number of function evaluations used overall in the main algorithm.

7. A version of Theorem 3.2 has already appeared in [1] without the dependency on $\rho$. So, the novel aspect of this lower bound is its dependency on $\rho$, which I found interesting. Please correct me if I'm wrong, but unfortunately, I cannot see the proof of this theorem in the paper.

8. I would like to know the authors' thoughts on higher-order smoothness: if we assume that the higher derivatives are uniformly bounded, do the authors believe it is possible to achieve the dependency of $d$ with respect to the dimension by estimating higher-order derivatives?

9. Could the authors specify the lower bound for $T$ for all the results in the paper to hold?


[1] Arya Akhavan, Massimiliano Pontil, and Alexandre Tsybakov. Exploiting higher order smoothness in derivative-free optimization and continuous bandits. Advances in Neural Information Processing Systems, 2020.

**Limitations:**

The authors adequately addressed the limitations.

---

> ### Author Rebuttal · Authors · 2024-08-06
>
> We appreciate the constructive feedback from the reviewer and have addressed each comment below.
>
> >  Why there is a need for two different gradient estimators?
>
> The two gradient estimators are tailored to the two phases of our algorithm, which serve distinct purposes. The first stage of our algorithm deals with potential initial conditions with unbounded gradients and highly skewed Hessians. Therefore, we need a gradient estimator whose variance does not depend on the gradient norm to ensure convergence. Thus, we employ the coordinate-wise estimator in BootstrappingEst instead of GradientEst (comparing Theorem 4.1 and 4.3).
>
> On the other hand, once the algorithm reaches a point sufficiently close to the global minimum, as specified in equation (6), the hyperellipsoid estimator in GradientEst provides a better bias-variance tradeoff (by a factor of $d$) as the third term on the RHS of inequality (3) becomes dominant. Therefore, in the second stage, we choose this estimator to reduce the estimation error.
>
> >  Why does the noise need to be sub-Gaussian?
>
> We would first like to emphasize that the sub-Guassian assumption we adopted in this work is merely to simplify the algorithmic design, allowing us to focus on presenting the key ideas for achieving minimax sample complexity. Even with general (non-sub-Guassian) noise distributions that have bounded variance, our results hold by employing more sophisticated mean-estimation methods (e.g., median-of-means instead of naive averaging). Therefore, the adoption of this assumption does not fundamentally weaken our results.
>
> However, the sub-Gaussian assumption becomes convenient in our setting compared to prior works, because we completely removed the Lipschitz gradient condition that those works assumed. Specifically, when the Lipschitz gradient condition is assumed, as shown in Akhavan et al. (2020), even simple gradient descent ensures that the squared norm of the updated gradient is linearly bounded by the squared norm of the previous gradient, allowing for a recursion based solely on the second moment of those gradients. Without this assumption, the squared norm of the updated gradient may depend on higher moments of the previous gradient.  This dependency is not immediately removed by Theorem 4.1 as it only states characterization up to the second moment. A sub-Gaussian gradient estimator can provide guarantees for these higher moments.
>
> > Motivate the second stage of the algorithm? Why they need a high probability bound in Theorem 4.3?
>
> Please refer to the response to the second question. The necessity of a high-probability condition similar to equation (6) in each step is fundamentally required due to the absence of the Lipschitz gradient assumption.
>
> In fact, we want to highlight that, even in the classical analysis of Newton's method, which assumes zero-error observations, the additional Lipschitz gradient condition was adopted to obtain non-trivial complexity bounds (e.g., see [A], Section 9.5.3), implying the non-trivialness of removing this smoothness condition even for achieving $O(1)$ expected simple regret with unbounded Hessian. We achieve this uniform convergence with a non-trivial modification of Newton’s algorithm. Therefore, we view both stages of our algorithm as main technical contributions.
>
> [A] Stephen P Boyd and Lieven Vandenberghe. Convex optimization. Cambridge university press, 2004.
>
> > where the authors used Assumption A (3)?
>
> The condition of A3 is not directly used in our algorithm. However, it is required for the regret analysis in the sense that the total number of samples needs to be greater than a function of R for the algorithm to enter the final stage, e.g, see Proposition E.1. This dependency only contributes to an additive term in the total number of samples, and hence does not change the sample complexity asymptotically.
>
> > discussion on the number of function evaluations.
>
> Thank you for the feedback. We will add discussions on the sample complexity analysis and show the total number of required function evaluations is indeed no greater than $T$.
>
> > Theorem 3.2 vs the lower bound in [1]?
>
> Due to space constraints, we have provided the proof of Theorem 3.2 in Appendix B. While we use different smoothness assumptions than those in [1], our lower bound can be considered a strengthened version of Theorem 6.1 in [1] under $\beta=3$, as discussed in Appendix A.
>
> > higher-order smoothness?
>
> We share the belief that under higher-order smoothness conditions, it is possible to achieve the corresponding optimal sample complexity with estimators of higher-order derivatives.
>
> > lower bound for 𝑇 for all the results in the paper to hold?
>
> For all our analysis, the required lower bound of $T$ are polynomials of $R$, $\rho$, $M$, and $d$, e.g., see the requirement in Proposition E.1. However, as the focus of this work is on the asymptotic analysis, we did not optimize our algorithm or analysis for these non-asymptotic requirements.

---

> > ### Comment · Reviewer_vYTV · 2024-08-08
> > **Rebuttal acknowledgment**
> >
> > I would like to thank the authors for the rebuttal. For now, I will maintain my current score. I plan to discuss the paper with the other reviewers and look forward to the author's discussions with them as well. I will update my score accordingly.

---

> > > ### Comment · Reviewer_vYTV · 2024-08-10
> > >
> > > Dear authors,
> > >
> > > After discussing with the area chair and other reviewers, I have maintained my score. Overall, I have a positive opinion of the paper and appreciate its technical novelty and contributions. However, as mentioned in my review and aligned with the opinion of reviewer cqSV, I believe the paper can be significantly improved in terms of writing and presentation. I recommend that many of the points you addressed in your response to the reviewers be included in the paper.
> > >
> > > Best regards,

---

### Decision · Program_Chairs · 2024-09-25

**Decision:**

Accept (poster)

**Comment:**

The paper addresses zero-order optimization of a strongly convex function with a Lipschitz continuous Hessian, proposing an algorithm that uses zero-order information to estimate the Hessian and gradient, enabling a second-order method to achieve minimax-optimal convergence for the optimization error, particularly improving the error's dependency on the dimension.

There was some initial concerns on the presentation of the paper, but these were offset by the strength of the results. The reviewers agree that the paper's contribution in bridging the optimality gap is significant. It corrects the current conjecture in the community that the optimal rate should be of the order $(d^2/T)^{2/3}$, as proposed by Akhavan et al. (2023). The idea of estimating the Hessian using zero-order information and employing a Newton-type method within this framework is novel and smart.

The authors are advised to revise the paper carefully to address the following three points:
* The paper could be significantly better written in terms of the motivations and intuitions behind the algorithms.
* The sub-Gaussian assumption is unnecessary, and the independence assumption is sufficient.
* The literature review is outdated.